



# A new method for operating a continuous flow diffusion chamber to investigate immersion freezing: assessment and performance study

Gourihar Kulkarni[1], Naruki Hiranuma[2], Ottmar Möhler[3], Kristina Höhler[3], Swarup China[1], Daniel J. Cziczo[4], Paul J. DeMott[5]

[1]Atmospheric Sciences and Global Change Division, Pacific Northwest National Laboratory, Richland, WA, USA
[2]Department of Life, Earth and Environmental Sciences, West Texas A&M University, Canyon, TX, USA
[3]Karlsruhe Institute of Technology (KIT), Institute of Meteorology and Climate Research (IMK-AAF), Eggenstein-Leopoldshafen, Germany
[4]Earth, Atmospheric and Planetary Sciences, Purdue University, IN, USA
[5]Department of Atmospheric Science, Colorado State University, Fort Collins, CO, USA

*Correspondence to*: Gourihar Kulkarni (Gourihar.Kulkarni@pnnl.gov)

**Abstract.** Glaciation in mixed-phase clouds predominately occurs through the immersion freezing mode where ice nucleating particles (INPs) immersed within supercooled droplets induce nucleation of ice. Currently, model representations of this process are a large source of uncertainty in simulating cloud radiative properties, and to constrain these estimates, continuous flow diffusion chamber (CFDC)-style INP devices are commonly used to assess the immersion freezing efficiencies of INPs. In this study, a new approach was explored to operating such an ice chamber that provides maximum activation of particles without droplet breakthrough and correction factor ambiguity to obtain high-quality INP measurements in a manner that has not been demonstrated as possible previously. The conditioning section of the chamber was maintained at ~-20°C and water relative humidity ($RH_w$) ~113% conditions to maximize the droplet activation, and the droplets were supercooled with an independently temperature-controlled nucleation section at a steady cooling rate (0.5°C min⁻¹) to induce the freezing of droplets and evaporation of unfrozen droplets. The performance of the modified ice chamber was evaluated using four INP species: K-feldspar, illite-NX, Argentinian soil dust, and airborne arable dust that had shown ice nucleation over a wide span of supercooled temperatures. Dry dispersed and size-selected K-feldspar particles were generated in the laboratory. Illite-NX and soil dust particles were sampled during the second phase of the Fifth International Ice Nucleation Workshop (FIN-02) campaign, and airborne arable dust particles were sampled from the aerosol inlet located on the rooftop of the laboratory. The measured ice nucleation efficiencies of model aerosols with a surface active site density ($n_s$) metric were higher, but mostly agreed within one order of magnitude compared to literature results.

## 1    Introduction

Atmospheric ice nucleation plays an important role in initiating precipitation in clouds that consist of a mixture of supercooled liquid water droplets and ice crystals and in catalyzing the formation of ice particles within high-altitude cirrus clouds (Lohmann and Feichter, 2006; Boucher et al., 2013). This important step toward ice formation also affects the lifetime and radiative properties of these clouds; however, ice nucleation mechanisms are poorly understood and parameterized in cloud



models (e.g., Hoose and Möhler, 2012; Murray et al., 2012; Kulkarni et al., 2012; Kanji et al. 2017; Knopf et al., 2018). Homogeneous ice nucleation is responsible for the formation of ice particles in dilute water and supercooled solution droplets at temperatures lower than ~-38°C (Pruppacher and Klett, 1997). Ice nucleation can also proceed through heterogeneous ice nucleation triggered by INPs (Vali et al., 2015). Multiple heterogeneous ice nucleation mechanisms have been proposed, such as deposition nucleation (ice formation on ice nucleating particles (INPs) directly from the vapor phase), contact freezing (freezing initiated by INPs the moment they come into contact with a supercooled droplet), and condensation and immersion freezing (freezing initiated by immersed INPs within the supercooled water or solution droplets). Nevertheless, the immersion freezing mode is thought to be the most important process toward the formation of ice particles within mixed-phase clouds (e.g., Ansmann et al., 2009; Westbrook and Illingworth, 2013).

Immersion freezing measurements are commonly made using continuous flow diffusion chamber (CFDC) devices (e.g., Rogers, 1988; Chen et al., 1998; Stetzer et al., 2008; Kanji and Abbatt, 2009; DeMott et al., 2010; Friedman et al., 2011; Chou et al., 2011; Jones et al., 2011; Kanji et al., 2013; Boose et al., 2016; Garimella et al., 2016, Schiebel, 2017; Zenker et al., 2017). These chambers consist of two ice-coated "parallel" walls held at different temperatures, and different ice supersaturations are achieved by regulating the temperature gradient. Aerosols sampled into CFDCs are subjected to known discrete temperature and relative humidity conditions, and when the water relative humidity ($RH_w$) is more than a few percents above 100%, droplets will activate on the majority of particles within the growth section of the chambers. CFDCs then typically have an evaporation section located at the bottom of the chamber where the wall temperatures are controlled in order to evaporate the droplets that did not freeze. Frozen droplets are counted using an optical particle counter (OPC) to determine the atmospheric INP concentrations. However, the maximum $RH_w$ values achievable in this manner can limit the ability to determine the maximum immersion freezing fraction (DeMott et al., 2015).

Here, we have expanded the capabilities of the CFDC-style device in order to achieve the maximum activation of particles to detect the immersion freezing number concentrations of INPs at various supercooled temperatures. We present data to assess the performance from a newly developed CFDC in which all individual aerosol particles are activated to droplets, and these droplets are exposed to a spectrum of supercooled temperatures. This is accomplished by modifying the existing design of the Pacific Northwest National Laboratory (PNNL) ice chamber (e.g., Friedman et al., 2011; Kulkarni at al., 2012). In the modified version, the growth section of the chamber was maintained at higher $RH_w$ conditions at moderate supercooling to activate all aerosol to supercooled droplets, whereas the evaporation section was always held at ice-saturated conditions and cooled over a range of temperatures at a known constant rate. The evaporation section serves two purposes in this case: it induces freezing of droplets and evaporates the unfrozen droplets. Validation experiments using standard salt solutions are presented. Various INP proxies of mineral dust types that have previously shown ice nucleation ability over a wide span of supercooled temperatures were used to test the performance of the modified chamber.

## 2    Experimental Design and Performance Validation


## 2.1 Description of the existing and modified chamber

The PNNL CFDC-style ice chamber operated in the traditional mode, referred to as Compact Ice Chamber (CIC)-PNNL, has been described in the literature (e.g., Friedman et al., 2011; Kulkarni at al., 2012). The chamber consists of two sections: a growth section and an evaporation section joined together but thermally insulated from each other. Each section consists of two parallel vertical surfaces that are both coated with a thin layer of ice, and these plates are independently temperature-controlled using external cooling baths (Lauda Brinkmann Inc.). Application of an ice layer (~0.5 mm thick) on these surfaces involved three consecutive steps: cooling the plates of the chamber to -25°C, filling the gap between the two parallel surfaces with deionized water (~18 MΩ cm), and expelling the water after 20 s. To produce the desired water- or ice-supersaturation conditions, a horizontal linear temperature gradient between the plates was applied, and the corresponding temperature and $RH_w$ or relative humidity with respect to ice ($RH_{ice}$) were calculated using the Murphy and Koop (2005) vapor pressure formulations. The sheath and sample flow rates were 5 and 1 liters per minute (LPM), respectively, resulting in a total particle residence of ~10 s in the chamber. The temperature gradient was applied such that supersaturation conditions $RH_w$ = ~106% were achieved in order to investigate the immersion freezing efficiencies of both atmospheric and laboratory-generated INPs. The OPC (CLiMET, model CI-3100) was used to classify the particles as ice crystals if they were greater than a certain size-threshold (~3 μm). The ice fraction ($F_{ice}$) was calculated by taking the ratio of the ice crystal concentration classified by the OPC to the total condensation nuclei (CN) concentration that entered the chamber. The CN concentration was provided by a condensation particle counter (CPC; TSI 3775). Blank experiments using dry and filtered sample air were also performed at the beginning and end of each experiment for ~10 minutes to calculate the background number of ice particles. Further, these ice particles were subtracted from the ice crystal concentration measured by the OPC, and the $F_{ice}$ was corrected.

Figure 1 shows a vertical cross-sectional geometry of the modified mode PNNL ice chamber. This chamber design has a parallel plate CFDC-style geometry, whose principle of generating a supersaturation between the two "parallel" surfaces and determining the $F_{ice}$ is similar to that of the existing CIC chamber, but with modifications as described here. The growth and evaporation sections of the CIC chamber are now referred to as conditioning and nucleation sections, respectively. The length of these two sections is identical (0.45 m), which limits the total residence time to ~10 s. The chamber wall temperature values as a function of time during one typical ice nucleation experiment are shown in Figure 2. During our study, the temperature controller of the cooling thermostat was programmed such that the warm and cold wall temperatures of the conditioning section were set to -9 and -27°C, respectively. Here, the choice of conditioning section temperature was based on previous knowledge that the onset temperature of the INP test species to induce nucleation of ice was at colder temperatures (< -20°C) and the lower detection limit to measure ice concentrations for temperatures warmer than -20°C. The shaded region shows the period (~30 minutes) of one ice nucleation measurement, i.e., the OPC data from this period only are analyzed. The ice fraction ($F_{ice}$) now indicates the cumulative fraction of droplets frozen as a function of decreasing temperature of the nucleation section (see S2). This metric of reporting ice nucleation results is commonly used to report frozen fraction vs. temperature ($F_{ice}$ vs. T) results (see e.g., DeMott et al., 2018; Kanji et al. 2017; Kohn et al. 2016). The isothermal conditions of the nucleation section



always help to maintain the ice saturation conditions and the complete section is cooled at a steady rate (0.5°C min⁻¹) by another separate cooling bath in order to determine the immersion freezing efficiency of INPs as a function supercooled temperature. The choice of steady-state cooling rate is empirical at this moment, and the experiment is terminated when the nucleation section reaches ~-44°C. The particle residence time (~5 s) and ice-saturated conditions of the nucleation section allow a sufficient size differential between supercooled droplets and ice crystals, and in fact, prevent "droplet breakthrough"

(Stetzer et al. 2008). While keeping the conditioning section conditions constant, the temperature of the nucleation section is raised to ~-20°C to prepare for the next ice nucleation measurement. This operation allows us to probe the immersion freezing efficiency of INPs at various temperatures (-20 to -44°C) multiple times (~ 5) before another layer of ice coating is applied. After more than five ice nucleation measurements or after approximately 3 hours, we see the reduced $F_{ice}$ of standard solution droplets (discussed below). The particle pulse experiments (Garimella et al. 2017) using size-selected 300 nm mobility

diameter ammonium sulfate particles and ~10.5 s pulse were performed. The temperature of the conditioning and nucleation sections were held constant at 20 °C, and the particle concentration at the chamber outlet using CPC was measured. These measurements show that ~16% of total particles that enter the chamber have moved outside of the lamina (Figure S1). Therefore, higher $RH_w$ values are utilized in the chamber to activate these particles to droplets (see below). At the entrance of the nucleation section, the temperature and $RH_w$ profiles can be unsteady, and to better understand the flow patterns of these

profiles within the transitioning zone, and its impact on droplet behavior, numerical simulations using computational fluid dynamics (CFD) are performed (as discussed below).

## 2.2 Numerical modeling

In this study, the warm and cold walls of the conditioning section were maintained at -9 and -27°C, respectively, and the nucleation section was maintained at -20°C. Analytical steady-state calculations based on Rogers (1988) were also used to

understand the nature of the flow velocity profile and the position of the aerosol lamina occupied between the warm and colder walls (Figure 3). The results show that the operating conditions of the chamber produce a skewed velocity profile and that the aerosol lamina is displaced toward the colder wall. The aerosol lamina is surrounded by filtered sheath flow, and its width is determined by the ratio of sample to sheath airflow. Because sample flow ideally occupies this fraction of the total flow, the aerosol lamina experiences a range of temperature and saturation conditions. The center temperature and $RH_w$ conditions,

including uncertainty across the aerosol lamina (assuming ideal confinement in the lamina), are ~-19.7 ± 0.7°C and ~113 ± 0.5%, respectively. Additional simulations are performed to understand the center temperature and $RH_w$ conditions required to confine the particles that are moved out of the aerosol lamina. We find that the revised uncertainties for center temperature and $RH_w$ conditions are ± 0.9°C and ± 0.7%, respectively.

CFD simulations are performed to achieve a complete description of the velocity, ice saturation, and temperature conditions

within the chamber (Figure 4a). A three-dimensional mesh of the chamber geometry was generated and exported to the commercially available CFD software ANSYS FLUENT 14.0 (2016). The CFD software solver was the pressure-based steady-





state Navier-Stokes equation with implicit and absolute velocity formulations. We used the RNG $\kappa - \varepsilon$ turbulence models, which treat velocity fluctuations better than other turbulence models for such geometry, and it was coupled with energy and viscous heating to enable the species transport model to better capture the effects of smaller eddies of fluid motion. The pressure

outlet boundary condition was used, as were the CFD solution method to couple the pressure-velocity, and the default SIMPLE scheme used. The Lagrangian discrete-phase model was used to simulate the potential INP trajectories released from the sample injection region to the outlet end of the chamber. The simulations were performed using an "uncoupled approach," which means the motion of INP particles does not influence the fluid flow pattern. The temperature and $RH_w$ fields of the INP trajectories were used to calculate the droplet growth and evaporation trajectories using a water vapor diffusion growth theory

(Rogers and Yau, 1988) that neglects temperature corrections and kinetic and ventilation effects and assumes perfect mass and thermal accommodation coefficients (S1).

The CFD simulated airflow velocity and $RH_{ice}$ profiles from the central region of the conditioning section are nearly similar to the analytical solution (Figure 2). Both calculations show the presence of maximum humidity values near the middle of the chamber but slightly displaced values toward the cold wall. The fluid flow temperature characteristics from the moment the

aerosol lamina joins the sheath flow show that the aerosol sample quickly (<0.5 s) cools at the entrance of the conditioning section. To gain a better understanding of $RH_w$ and temperature conditions within the conditioning and nucleation sections, the simulated data set of a potential INP trajectory transiting within the chamber is shown (Figure 4b). The potential INPs experience nearly constant $RH_w$ and temperature conditions within a short time, ~1 s, after entering the conditioning section. The potential INPs are assumed (i.e., sub-saturated particle growth is ignored) to activate to droplets because they are greater

than cloud condensation nuclei sizes (Seinfeld and Pandis, 2016) and grow as long as $RH_w$ is increasing or remains constant. As the INPs enter the nucleation section, their $RH_w$ and temperature values equilibrate with the nucleation section conditions. These calculations show that the droplets grow to ~4 µm in diameter, and they shrink as $RH_w$ and temperature decrease (within the nucleation section). Note that droplet freezing within the nucleation section is not simulated in these simulations. Additional simulations with the nucleation section temperature set to -30°C and -37.5°C were also performed (Figure S2-5). These

simulations show that the $RH_w$ field of a potential INP slightly decreases (~0.5%) and then increases within a very short period of time (<0.5 s) at the entrance region of the nucleation section. However, calculations show that such a perturbation does not affect the droplet evaporation behavior within the nucleation section as they all evaporate within ~1s after they enter the nucleation section and within uncertainty limits of set temperature of the nucleation section, but not before they reach the set temperature (see Figures S2-4). The simulations are extended to understand the $RH_w$ and temperature conditions of potential

INPs released from different regions of the inlet section of the chamber. Simulations of five potential INPs are shown in S5. It is observed that INPs experience various temperature conditions (-17 to -19.5°C) within the conditioning section, however, after ~0.5s they all enter the nucleation section the temperature of each trajectory is identical. Additional evaporative cooling calculations are performed to understand the suppression of droplet temperature while they are entering the nucleation section. In the nucleation section the supercooled droplets experience sub-saturation ($RH_w > 0.8$) and colder temperature conditions (>



-37.5°C). The Kulmala evaporative model (Su et al. 2018) was used to determine the surface temperature of these droplets using steady-state aerosol lamina airflow velocity (Figure 3) and theoretical predicted $RH_w$ fields (Figures S2-4). The calculations (Table S1) show the negligible effect of evaporative cooling on the droplet temperature such that additional supercooling is within the reported temperature uncertainty (= ± 0.7°C) across the aerosol lamina, and therefore droplet evaporating cooling effects within the nucleation section are ignored.

## 170    2.3     Homogeneous freezing of ammonium sulfate particles

The temperature conditions within the nucleation section were validated using size-selected ammonium sulfate (AS) particles. These particles were generated by atomization of an aqueous solution made by dissolving AS (1 g) and Milli-Q water (18.2 MΩ cm; 100 g) resulting in a 1 wt% solution concentration using a constant output atomizer (TSI 3076). The atomized droplets were transported through a diffusion drier to obtain the dry particles, which were further transported to the differential mobility

analyzer (DMA; TSI 3081) to obtain size-selected particles that had mobility diameters of 200 nm. The concentration of these size-selected particles was measured using a CPC, and the particles were further transported to the ice chamber. As stated previously, the temperature and $RH_w$ conditions within the conditioning section were -20°C and ~113%, respectively, and these conditions were held constant, which led to droplet activation of size-selected AS particles. Next, the nucleation section was steadily cooled from -20 to -40°C, and the ice particles exiting the chamber were classified as ice particles. The ice particle

size distribution with supercooling is shown in Figure 5a. The results show that the droplets began to freeze via a homogeneous freezing mode at ~-37.5°C. The maximum number of ice particle concentrations was observed at ~-38.5°C when all the droplets froze. The nucleation section is always maintained at $RH_{ice}$ = 100% (see Figure 4a), and such ice saturation conditions either do not grow or sublimate the ice crystals. Therefore, ice particle size measured by the OPC can be representative of the size of the droplet while freezing. At slightly warmer temperature (between -38.5 and -37.5 °C), we observe ice particles of

size ~ 2.0 µm. The appearance of these smaller ice crystals could be because of the freezing of these smaller droplets (a consequence of evaporation within the entrance zone of the nucleation section) compared to ~ 5.0 µm droplets at ~-38.5°C. These homogeneous freezing threshold temperature values are in agreement with previous studies (e.g., Ignatius et al., 2016; Kohn et al., 2016). For example, Kohn et al. (2016) found 100% freezing of the supercooled dilute aqueous solution droplets at ~-38.2°C. Theoretical calculations using a homogeneous nucleation rate (e.g., Earle et al. 2010; Atkinson et al. 2016) were

performed to predict the homogeneous freezing curves of the droplet of size 4 µm in diameter. Homogenous freezing curves for various probable droplet residence times within the nucleation section are shown in Figure S6. We find good agreement between the experimental and predicted freezing temperatures, and the freezing results (see section 3) at warmer temperatures (> -37°C) can be ascribed as the heterogeneous freezing of the droplets or immersion freezing. Our results also show the complete evaporation of supercooled droplets within the nucleation section, because no ice particles are observed above

~-37.5°C.



This experimental setup was further applied to understand the relationship between the $F_{ice}$ of AS particles relative to the $RH_w$ conditions within the conditioning section. The aim was to investigate the $RH_w$ value at which all the size-selected AS particles activate to droplets. Here, the nucleation section was held at -42°C to induce homogeneous freezing of solution droplets, while the $RH_w$ within the conditioning section was steadily increased. It can be seen that $RH_w$ values close to 113% are required

before all the AS particles are activated to droplets (Figure 5b). Higher $RH_w$ values enable the encapsulation of all particles that are within and may spread outside (Garimella et al. 2017) the width of aerosol lamina into droplets, but high saturation conditions also help to grow the droplets to the larger size; so, they survive long enough to induce the freezing of droplets within the nucleation section.

### 2.4     Sample preparation

The immersion freezing efficiency of K-feldspar, illite-NX, Argentinian soil dust, and airborne arable dust particles was measured to test the performance of the ice chamber operated in a new mode. K-feldspar (BCS376) was purchased from the Bureau of Analysed Sampled Ltd, UK. Dry dispersed (TSI 3433) K-feldspar particles that had a mobility diameter of 400 nm were size-selected by a DMA, and these nearly monodisperse particles were transported to the CPC and ice nucleation chamber. Based on theoretical calculations (Baron and Willeke, 2001), the distribution of these classified particles may also

contain sub-populations of double (~700 nm) and triple (~985 nm) charged particles. Laboratory measurements showed that the contribution of double and triple charged particles was less than 7 and 3%, respectively. Therefore, the multiply charged particle contribution is neglected, and the K-feldspar aerosol stream is assumed to consist only of particles whose mobility diameter equals 400 nm. However, the surface area of multiple charged particles could influence $F_{ice}$, because these large particles (>400 nm) provide larger surface areas (Lüönd et al., 2010). Illite-NX and Argentinian soil dust were sampled at the

AIDA (Aerosol Interaction and Dynamics in the Atmosphere) chamber facility during the Fifth International Ice Nucleation Workshop (FIN-02) campaign (DeMott et al., 2018). During the campaign, the two aerosol types were dry dispersed in two different chambers: an 84 m³ AIDA chamber and a 4 m³ aerosol particle chamber (APC); but in this study, we sampled directly from the APC. The details of particle generation and aerosol properties are described by DeMott et al. (2018). The direct sampling of these two aerosol types corresponds to experiment numbers 8 and 10 on 3/16/2015 and 3/17/2015, respectively.

Airborne arable dust particles were sampled at the PNNL sampling site during a regional windblown dust event. The PNNL sampling site is located within the Columbia Plateau, WA, the USA, which is confined by the Rocky Mountains to the east, the Blue Mountains to the south, and the Cascade Mountains to the west. The region once was covered with basalt lava, but now is built up with loose topsoil – loess. This fine soil, which is erodible, and the agricultural dryland farming practices make this dry soil susceptible to wind erosion. The sampling was performed during one dust event on 5/11/2017, and the average

temperature, humidity, and wind speed during this day were 18°C, 60%, and 14 mph, respectively. The sampling port was ~9 m above the ground on the rooftop of the Atmospheric Measurements Laboratory located on the PNNL campus in Richland, WA. The airborne dust particles were drawn into the laboratory through a cyclone impactor (URG-200-30EH), which was operated at 30 LPM to obtain a cut point diameter equal to 1.5 µm. This size-selective sampling allowed for removal of the



larger particles (>1.5 µm) and therefore helped to classify unambiguously the ice crystals larger than 3 µm using an OPC. The

CN concentration of airborne arable dust particles (>0.1 µm) was measured using a laser aerosol spectrometer (LAS; TSI 3340). The $F_{ice}$ was calculated by determining the ratio of ice crystals provided by the OPC to the CN counts measured by the LAS. To better understand the size distribution and composition of these airborne dust particles, the particles were collected on a carbon type-B film (Ted Pella Inc.; 01814-F) for scanning electron microscopy-energy dispersive x-ray spectroscopy (SEM-EDS) analysis. The films were mounted on the C-and D-stages of a SKC Sioutas impactor that had 50% cut-points of

0.5 and 1.0 µm, respectively. The impactor was operated at 9 LPM, and a total of 1183 particles were analyzed. Figure S7 shows the exemplary SEM images. The images reveal that the particles are mostly composed of minerals, and the size distribution shows the mean area equivalent diameter of ~0.53 µm.

## 3    Results and Discussion

The modified ice nucleation chamber was operated to measure the maximum immersion freezing fraction of INPs. The

modified design allowed for the faster (~ 30 minutes) accumulation of immersion freezing data points to develop a continuous representation of the immersion freezing behavior of INPs compared to the traditional CIC-PNNL design, where immersion freezing was investigated at discrete temperatures. These expanded capabilities were demonstrated by measuring the immersion freezing properties of four INP substances, including K-feldspar, illite-NX, Argentinian soil dust, and airborne arable dust particles.

The measurements of immersion freezing properties of the four samples were investigated at temperatures between -20 and -38°C. The averaged $F_{ice}$ data over $\Delta T = 0.25$°C temperature intervals were plotted against the midpoint temperature of each bin (Figure 6). The vertical and horizontal error bars are equal to the one standard deviation of the $F_{ice}$ measurements (n = 3) and temperature uncertainty (± 0.4°C) across the nucleation section, respectively. Freezing experiments with AS solution droplets show the homogeneous freezing threshold temperature conditions below ~-38°C, and therefore $F_{ice}$ data points above

this temperature can be attributed to the immersion freezing mode only. Figure 6 shows that four INP materials exhibit a distribution of immersion freezing temperatures. The $F_{ice}$ of all INP species increased with decreasing temperature consistent with many past studies (e.g., Kanji et al., 2017). The droplets containing immersed K-feldspar particles froze at higher temperatures. The median freezing temperatures (i.e., the temperature at which 50% of the droplets froze) of K-feldspar, illite-NX, Argentinian soil dust, and airborne arable dust particles was -25.4, -32.6, -31.4, and -31.8°C, respectively, and the

difference between freezing temperatures corresponding to $F_{ice}$ equal to 90% and 10% was approximately between ~4.5 and 7.5°C for all four INP materials.

Additional experiments were performed to confirm that the dynamic temperature conditions (steady-state cooling) of the nucleation section does not affect the freezing behavior of particles. The measurements were conducted on K-feldspar and airborne arable dust particles that were prepared as described above in the sample preparation section. The temperatures of the

warm and cold walls of the conditioning section were maintained at -9 and -27°C, respectively, and the nucleation section





temperature was held constant (instead of steady-state cooling). The immersion freezing fraction data points of these two species are shown as solid symbols in Figure 6. A good agreement with the results obtained where the chamber was operated in a new mode was observed, which suggests that the temperature ramping operation of the nucleation section (0.5°C min$^{-1}$) does not affect the performance of INP activation experiments.

These $F_{ice}$ measurements were further analyzed using the ice nucleation active site density ($n_s$) approach that allowed to compare against other studies (see below). This approach also allowed us to compare results directly with literature data obtained using different experimental setups and various direct and post-processing INP instruments and particle generation methods. The $n_s$ indicates the cumulative number of ice active sites that are present per unit area of particle surface, and that induce nucleation of ice upon cooling from 0°C to experimental temperature T. In this calculation, time-dependence is

neglected, and it is assumed that the different active sites present within the droplets are responsible for the nucleation of ice. The $n_s$ calculation follows DeMott et al. (2018) and Hiranuma et al. (2015):

$$n_s(T) = \frac{-\ln(1 - F_{ice})}{A} \approx \frac{F_{ice}}{A} \tag{1}$$

where $A$ is the surface area per particle. For K-feldspar and airborne arable dust particles analysis, the surface area is calculated

assuming the particles are spherical, and this assumption may overestimate the $n_s$; therefore, calculations should be viewed as the upper estimates of $n_s$. The size distribution and CPC concentrations were used to calculate the $A$ of individual airborne arable dust particles, as described by Niemand et al. (2012). For illite-NX and Argentinian soil dust particles, the $A$ was obtained from the FIN-02 data archive (DeMott et al., 2018). The error in $n_s$ (Eq. 1) was calculated using the error propagation method based on the uncertainties of the $F_{ice}$ and $A$.

Figure 7 shows $n_s$ for the four INP materials tested in this work in comparison to parameterizations reported in previous studies. $n_s$ for K-feldspar is compared to the fit published by Atkinson et al. (2013). There is a good agreement with our measurements for temperatures warmer than -26°C. Atkinson et al. (2013) used a droplet-freezing cold stage technique, where a known amount of K-feldspar material was present in each droplet sized between 14 and 16 μm. These droplets were cooled at a rate of 1°C min$^{-1}$, and droplet-freezing temperature data were used to construct the $n_s$ parameterization. Note that the $n_s$ fit from

Atkinson et al. (2013) is valid up to -25°C. In our work, we extrapolated the fit outside this limit to colder temperatures for comparison. However, such linear extrapolation to colder temperatures may not be correct, because, as both Niedermeier et al. (2015) and DeMott et al. (2018), the latter from the FIN-02 campaign, have shown, the $n_s$ values level off at temperatures colder than -25°C. $n_s$ for airborne arable dust was compared with the previous studies. Niemand et al. (2012) derived the $n_s$ fit using combined immersion freezing data from various natural dusts (Asian soil dust, Canary island dust, Saharan dust, and

Israel dust). Recently, Ullrich et al. (2017) developed $n_s$ parameterization using immersion freezing $n_s$ densities of various arable dusts (Saharan desert dust, Asian desert dust, Israel desert dust, Canary Island dust) for the temperature range from -14 to -30°C. Tobo et al. (2014) investigated the INP abilities of agriculture soils dusts collected from Wyoming, USA. Boose et al. (2016) investigated the INP efficiencies of airborne dust samples from four locations (Crete, Egypt, Peloponnese, and





Tenerife) and generated the minimum to maximum bounds of $n_s$ from -29 to -37°C. The comparison of our results with these

previous results shows good agreement within one order of magnitude at colder temperatures, but the data diverge at warmer

temperatures. This could be the consequence of a particularly active soil dust present in the local region. $n_s$ for illite-NX is

compared to Hiranuma et al. (2015), who combined immersion freezing data from several direct processing INP methods to

develop a $n_s$ parameterization. Here, we used the Gumbel cumulative distribution linear fit parameters derived from dry

dispersion measurements to generate the $n_s$ fit. The present data agree within one order at warmer (-28 to -30°C) and colder

temperatures (-34 to -38°C), but at other temperatures ( -30 to -34°C) the data diverge. Finally, we compared our data with $n_s$

parameterization from Steinke et al. (2016). Steinke et al. (2016) used immersion freezing data from four soil dust samples

(Mongolian soil, Karlsruhe soil, German soil, and Argentinian soil) to produce a $n_s$ fit that is valid over a temperature range

between -26 to -11°C. We extrapolated the $n_s$ fit toward colder temperatures, and comparison shows higher $n_s$ values but

overlaps within the order of magnitude with others.

Figure 7 (a, c, and d) also shows the $n_s$ results reported by five different direct processing INP instruments used in the FIN-02

campaign (DeMott et al., 2018). Our data for K-feldspar nearly align with the others at warmer temperatures (> -28°C). For

the illite-NX sample, agreement with the PIMCA-PINC method is within one order of magnitude, but the agreement is

observed within two orders of magnitude with others. The present data for Argentinian soil dust aligns with the PIMCA-PINC

method and agrees with the others within one order of magnitude. The discrepancy between present and others could be

attributed to the different capabilities employed by individual measurement methods to investigate the immersion freezing

properties. Previously evaporative freezing by contact nucleation inside-out has been hypothesized to explain the higher

freezing temperatures and rates of ice formation observed during droplet evaporation (Durant and Shaw, 2005). Durant and

Shaw (2005) showed that water droplets containing individual insoluble INPs freeze at a higher temperature compared to the

immersion freezing mechanisms. We cannot rule out that the evaporative freezing mechanism may be occurring in our

experiments, and it would be responsible for the higher ns values compared to other studies. The comparison with the CIC-

PNNL chamber showed that present data agree within one order of magnitude. Note that CIC-PNNL (PNNL ice chamber but

operated in a traditional mode; Friedman et al., 2011; Kulkarni et al., 2012) was operated at $RH_w$ = 106%, and its operation

limited investigating immersion freezing on the entire particle population. It can be observed that for illite-NX and Argentinian

soil dust samples a correction factor of 4 up to 5 is needed to apply to the CIC-PNNL data to match with the data from the new

mode of chamber operation.

## 4     Conclusions

An alternative method of operating a CFDC-style ice chamber was explored to detect the immersion freezing ability of INPs.

This new mode of operation allowed us to obtain maximum immersion freezing fractions of INPs without droplet breakthrough

ambiguity. Here, instead of investigating immersion freezing at discrete temperatures, immersion freezing was investigated by

activating particles to droplets at high $RH_w$ followed by steady cooling under imposed ice-saturated conditions. The chamber





performance was evaluated by testing the ice nucleation ability of four INP materials: K-feldspar, illite-NX, Argentinian soil dust, and airborne arable dust particles. In addition, we performed CFD simulations to evaluate flow, humidity, and temperature performance. The results indicate that these three thermodynamic conditions are locally fully developed, which confirms constant mass and thermal flux, and therefore steady operating conditions within the chamber. Tests using size-selected AS particles showed that homogeneous freezing of solution droplets occurs in agreement with theory and previous study results, and that to activate all the particles to droplets high $RH_w$ values of ~113% are needed. Analytical and CFD calculations indicate that such high values are needed to grow the droplets to larger sizes so that they can survive long enough to induce freezing and to allow the particles that may have escaped the aerosol lamina to activate into droplets. Tests using the four INP materials demonstrated the activation of all individual particles to generate immersion freezing spectra in terms of $F_{ice}$ and $n_s$. Experimental results indicate that K-feldspar minerals induced detectable ice formation at ~-22°C and maximum $F_{ice}$ (= 90%) was observed at -28°C. The other three samples induced nucleation of ice at temperatures colder than -26°C, and their maximum $F_{ice}$ (= 90%) was observed ~-36°C. The $F_{ice}$ was normalized using particle surface area to calculate the $n_s$, and these $n_s$ calculations show that our results are comparable to the parameterizations and data reported in the literature. We find that the majority of our $n_s$ results are higher within one order of magnitude than others. Analysis of such high temporal resolution immersion freezing measurements could offer better insights into the freezing properties of INPs, thereby moving us toward improved representations of the immersion freezing ability of INPs for cloud models.

*Data availability.* Data plotted in this paper are available upon request.

*Author contribution.* GK analyzed the data and wrote the paper. NH, OM, KK, SC, DC and PJD contributed and commented on all results. SC provided airborne arable dust composition and morphology results.

*Competing interests.* The authors declare that they have no conflict of interest.

*Acknowledgments.* The work was supported by the Office of Science of the U.S. Department of Energy (DOE) as part of the Atmospheric System Research Program. We thank microscopy capability for performing SEM-EDX analysis at Environmental Molecular Sciences Laboratory, which is a national scientific user facility located at PNNL in Richland, Washington. PNNL is operated for the U.S. DOE by the Battelle Memorial Institute under contract DEAC05-76RL0 1830. The FIN-02 campaign was partially supported by the U.S. National Science Foundation grant no. AGS-1339264, and by the U.S. DOEs Atmospheric System Research, an Office of Science, Office of Biological and Environmental Research program, under grant no. DE-SC0014487. Paul J. DeMott acknowledges additional support from the U.S. National Science Foundation award numbers 1358495 and 1660486. We also acknowledge support from the AIDA team and organizers of FIN-02 campaign.





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





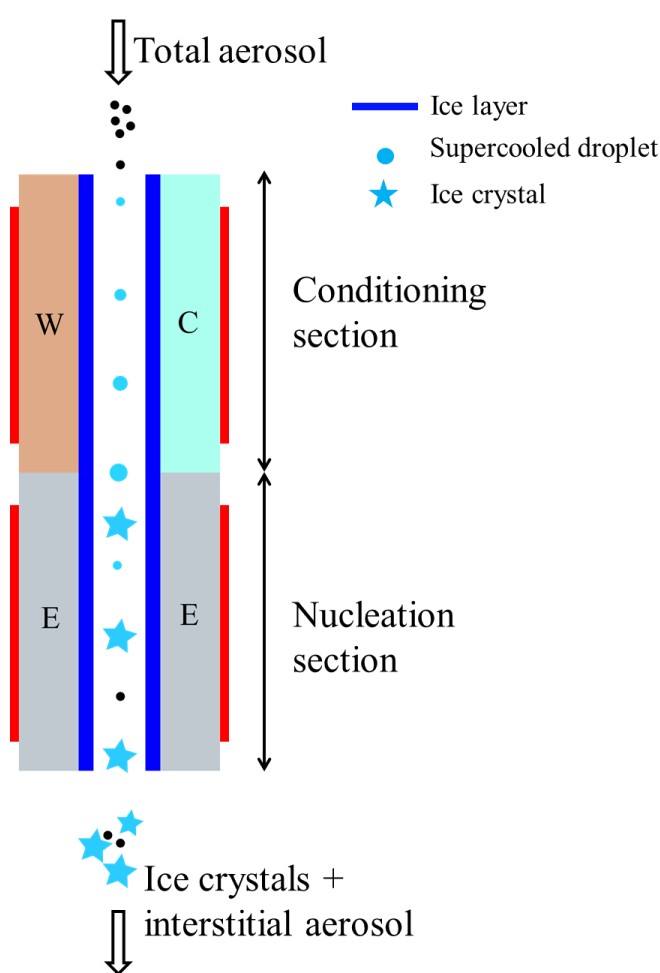

**Figure 1: Schematic showing the geometry of the modified ice chamber (expanded for clarity). INP proxies are activated to droplets within the conditioning section, and these supercooled droplets are steadily cooled within the nucleation section, from ~-20°C to ~-42°C to induce freezing of droplets and evaporate unfrozen supercooling droplets. The residence time in each section of the chamber is ~5 s, and the ice layer spans both sections of the chamber. A cyclone impactor upstream of the ice chamber is used to remove the larger particles (>1.5 µm in diameter) while sampling airborne arable dust particles. The heating tapes (red rectangular strip) are attached to the walls to precisely control the temperature of the walls. W –warm wall; C – cold wall; E – Nucleation section wall.**






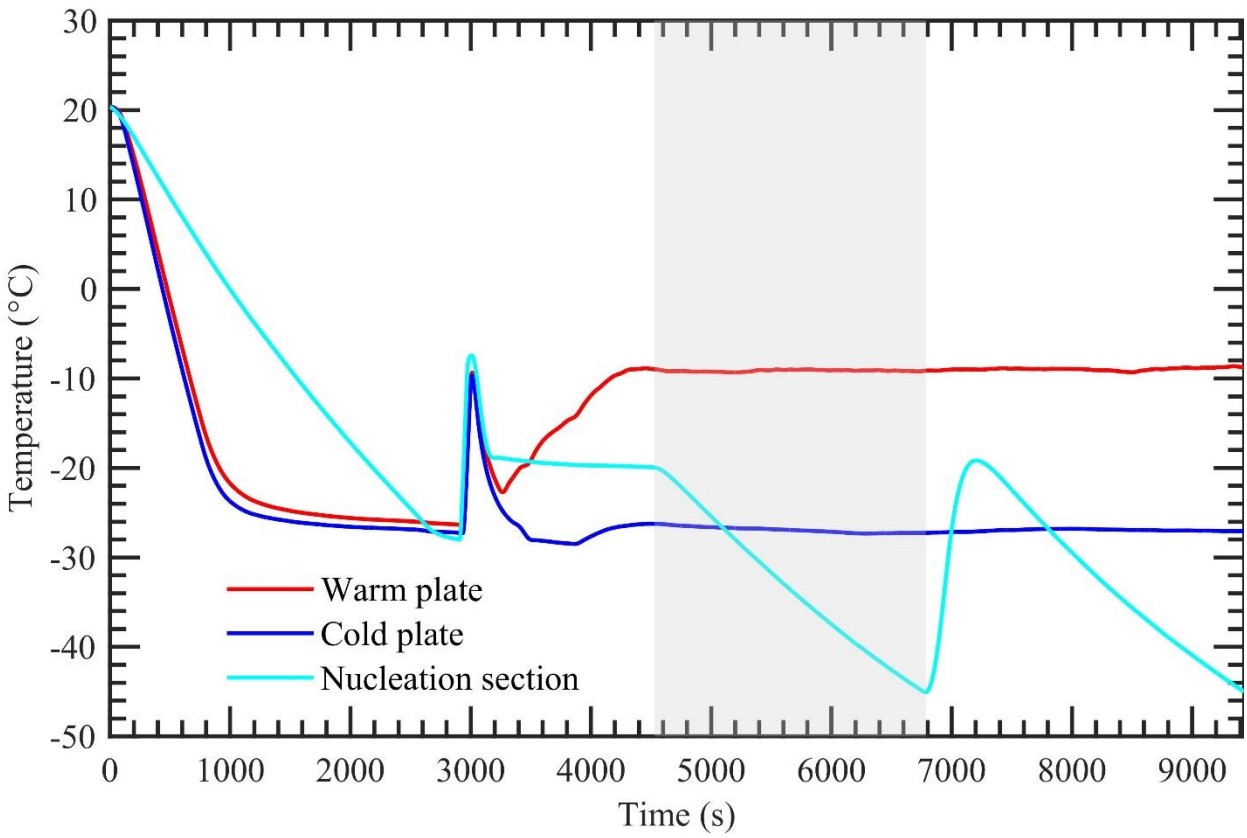

**Figure 2: Measured temperature of warm, cold, and nucleation section walls during a typical experiment. The shaded area indicates the experimental conditions during one ice nucleation measurement. During this INP measurement, the temperature of both warm and cold walls is kept constant, while the nucleation section is cooled at a steady rate (0.5°C min⁻¹).**




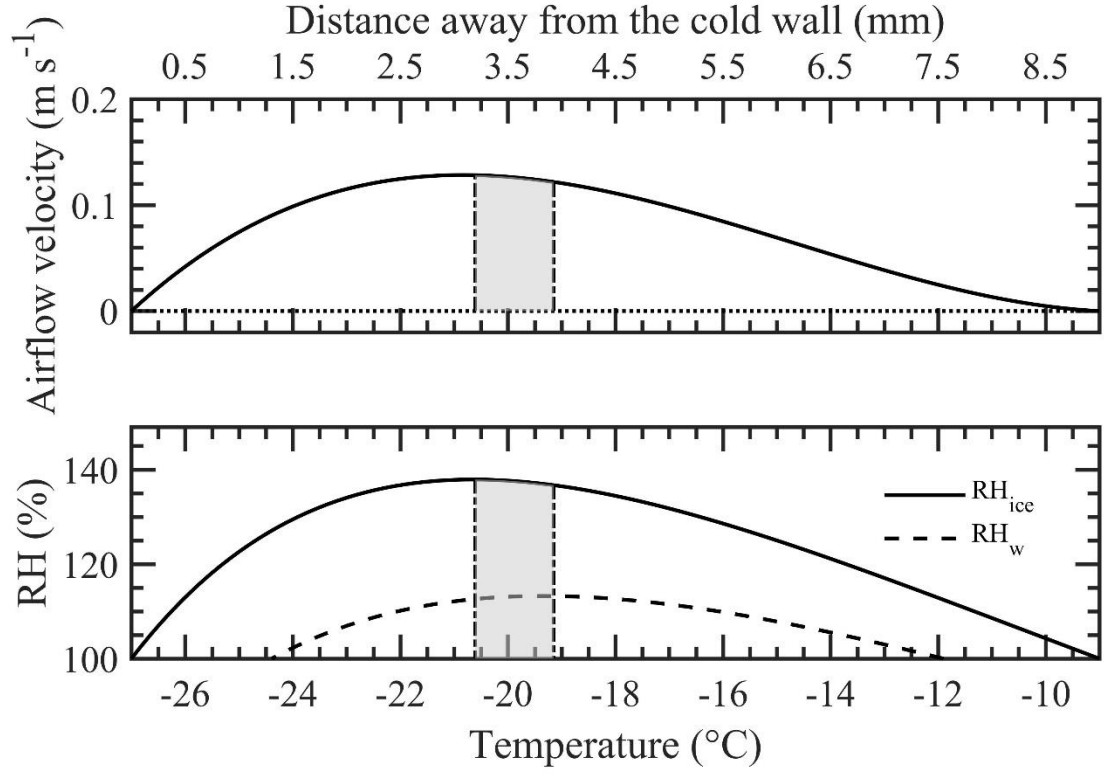

Figure 3: **Steady-state airflow velocity and relative humidity (RH) conditions calculated using the mathematical model developed by Rogers (1988) within the conditioning section of the ice chamber. The chamber warm wall (left) and the cold wall (right) are at -9 and -27°C, respectively. The shaded area between the two vertical dashed-dotted lines shows the boundaries of aerosol lamina under at these above temperatures and flow conditions (sheath flow: 5 LPM and sample flow: 1 LPM). The profiles are asymmetric because of the thermophoretic drift of the flow, caused by the thermal gradient between the walls, towards the colder wall. The conditioning section is always supersaturated with respect to ice ($RH_{ice}$>100%), and except the near-wall positions, the section is also supersaturated with respect to water ($RH_w$>100%).**




(a)

Velocity (m/s)   RH$_{ice}$   Temperature (°C)





(b)

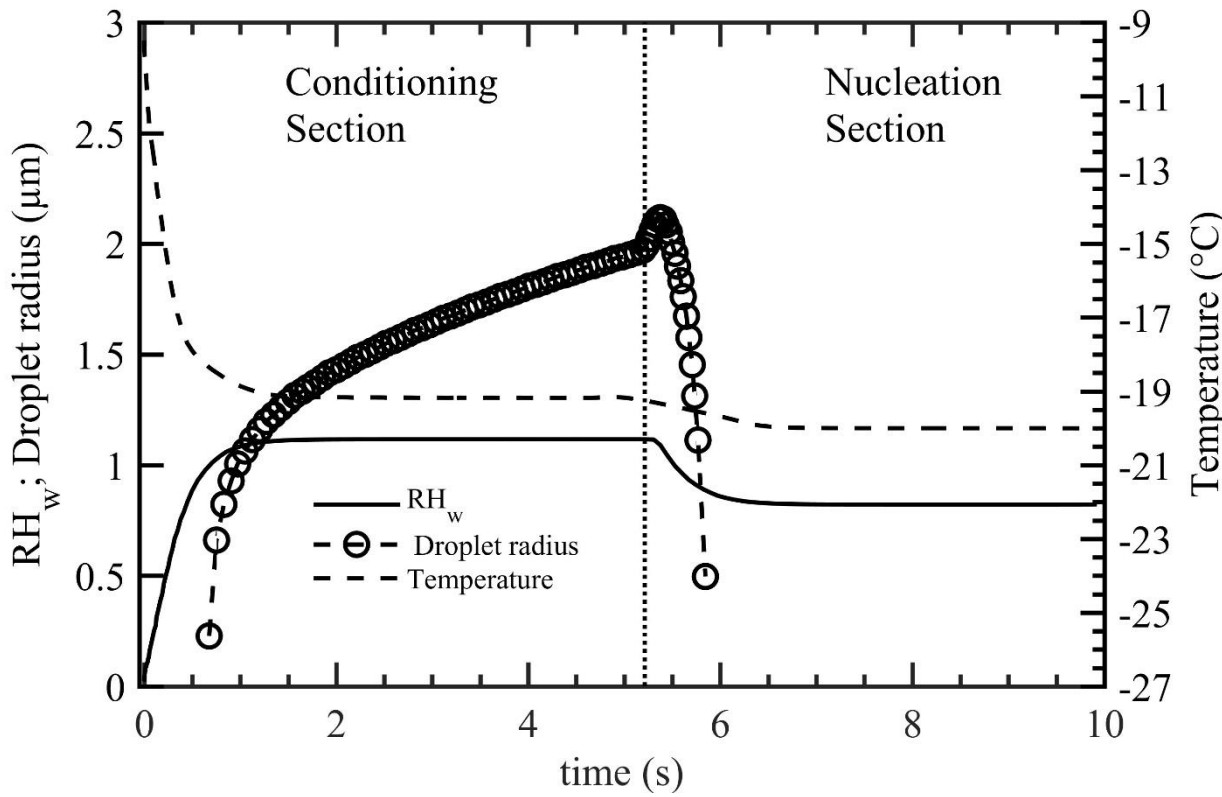

**Figure 4: (a) Contours of CFD calculated airflow velocity, $RH_{ice}$, and temperature profiles within the ice chamber. Warm and cold walls of conditioning section are maintained at -9 and -27°C, respectively. The nucleation section is maintained at -20°C. The dashed line shows the trajectory of a single INP within the aerosol lamina transiting through the chamber. (b) CFD calculated temperature and $RH_w$ profiles of a potential INP released from the sample injection region to the outlet end of the chamber. Analytical calculations**
**of droplet growth and evaporation of such a potential INP (0.3 µm in diameter) are also shown. The left and right sides of the vertical dotted line represent the conditioning and nucleation sections, respectively. See the text for more details. Simulations results at other nucleation section temperatures are shown in Fig. S1-4.**







(a)

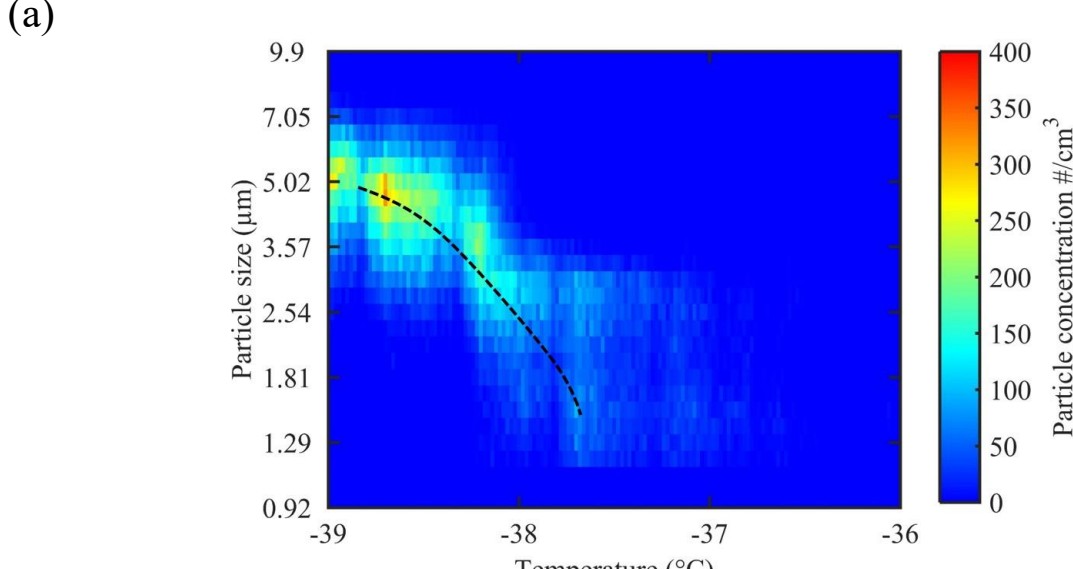

(b)

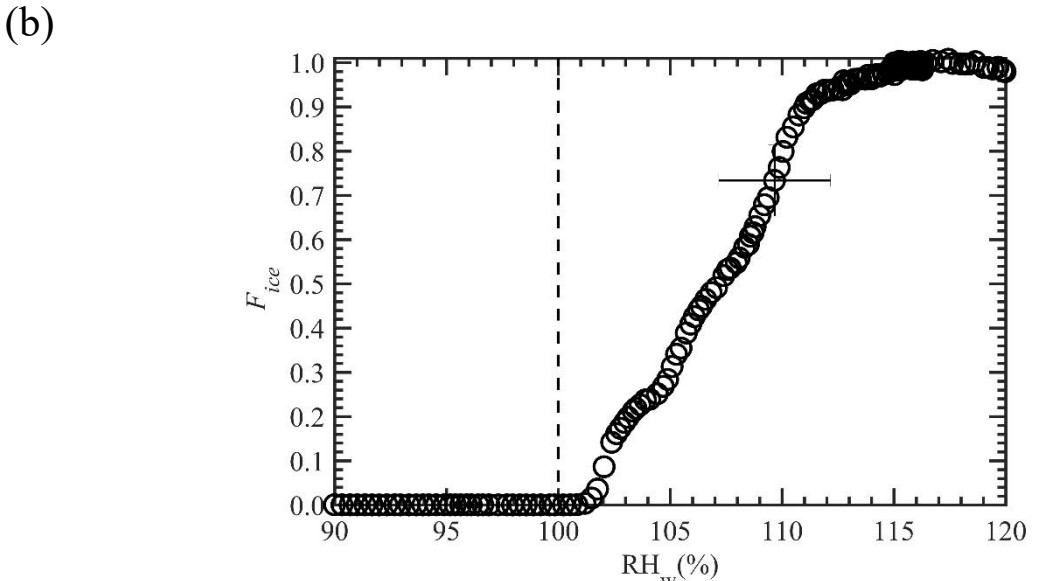

**Figure 5: Homogenous freezing of water droplets containing one wt. % ammonium sulfate solution. (a) OPC classified ice particle**
**concentrations as a function of ice crystal diameter at different temperatures. Warm and cold walls of conditioning section are**
**maintained at -9 and -27°C, respectively. (b) The fraction of frozen solution droplets with $RH_w$, where the temperature of the**
**nucleation section is maintained constant at -42°C to induce droplet freezing via the homogeneous freezing mode and $RH_w$ within**
**the conditioning section was steadily increased from 90 to 120%. Slightly colder temperature (-42°C) than homogeneous freezing**
**limit (~-38.5°C; panel a) is used to account for the uncertainty within temperature and $RH_w$ conditions. The dashed line in panel (a)**
**and (b) indicate the increase in freezing fraction of droplets trend (for illustration purpose) and the onset of saturation line,**



respectively. The uncertainty in $RH_w$ is shown as an error bar (see the text for more details). The uncertainty in $F_{ice}$ is one standard deviation (n = 3). For clarity, error bars are shown only for one data point.







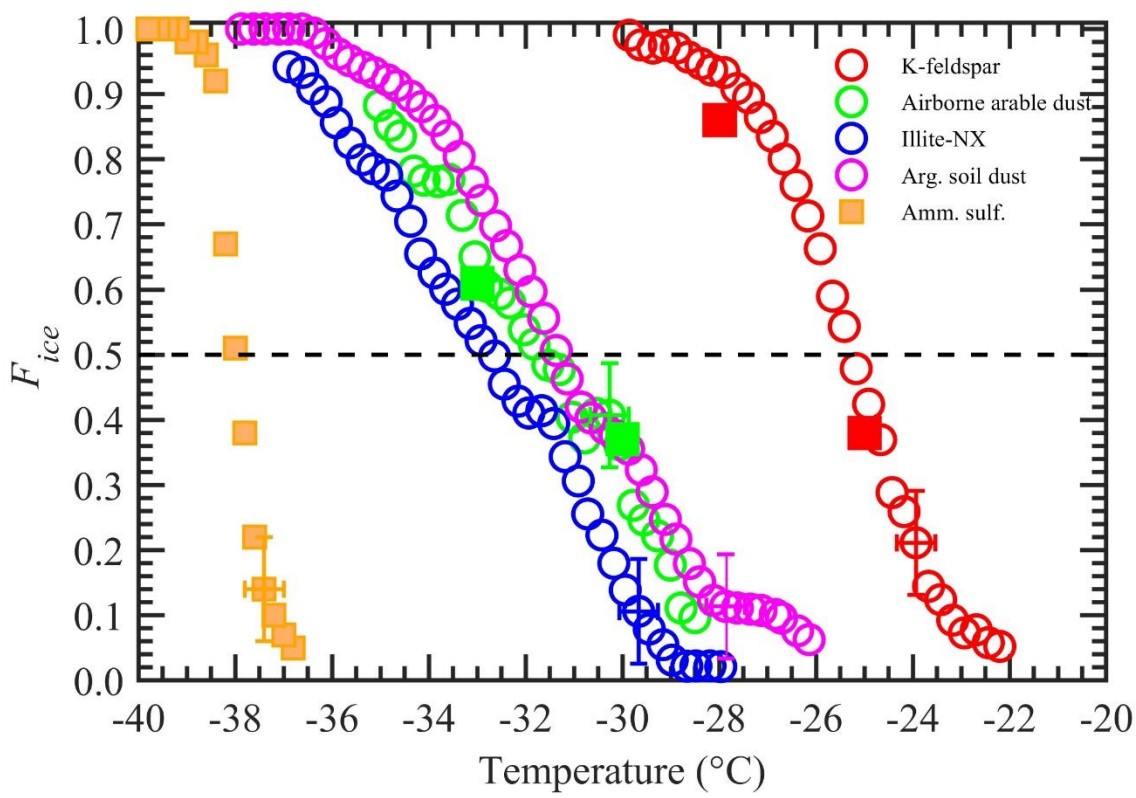

**Figure 6: The $F_{ice}$ of four INP test species as a function of temperature. The vertical error bar represents the one standard deviation of the three repeat experiments (n = 3). Temperature measurements had ±0.4°C uncertainty. For clarity, error bars are shown for only one data point. Orange solid square markers represent the freezing temperatures of water droplets containing one wt. % AS. Other solid square markers represented the data points when the chamber was run in a mode where the evaporation section was operated at a steady-state temperature (instead of steady cooling). The horizontal dashed line represents 50% $F_{ice}$.**




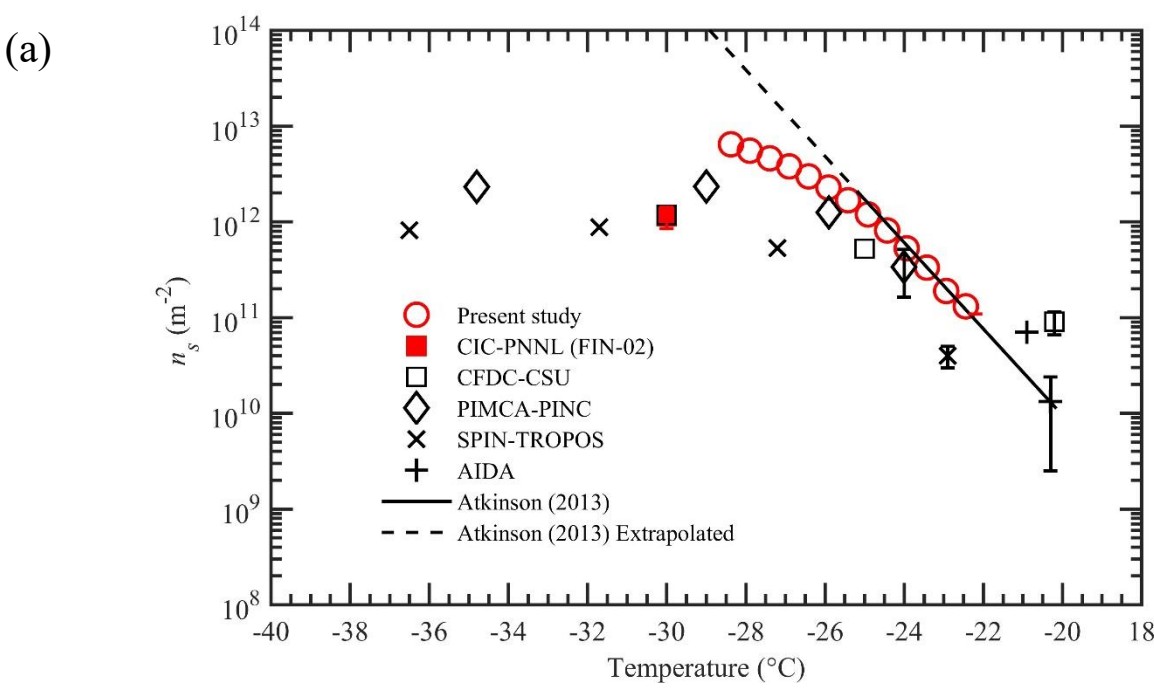

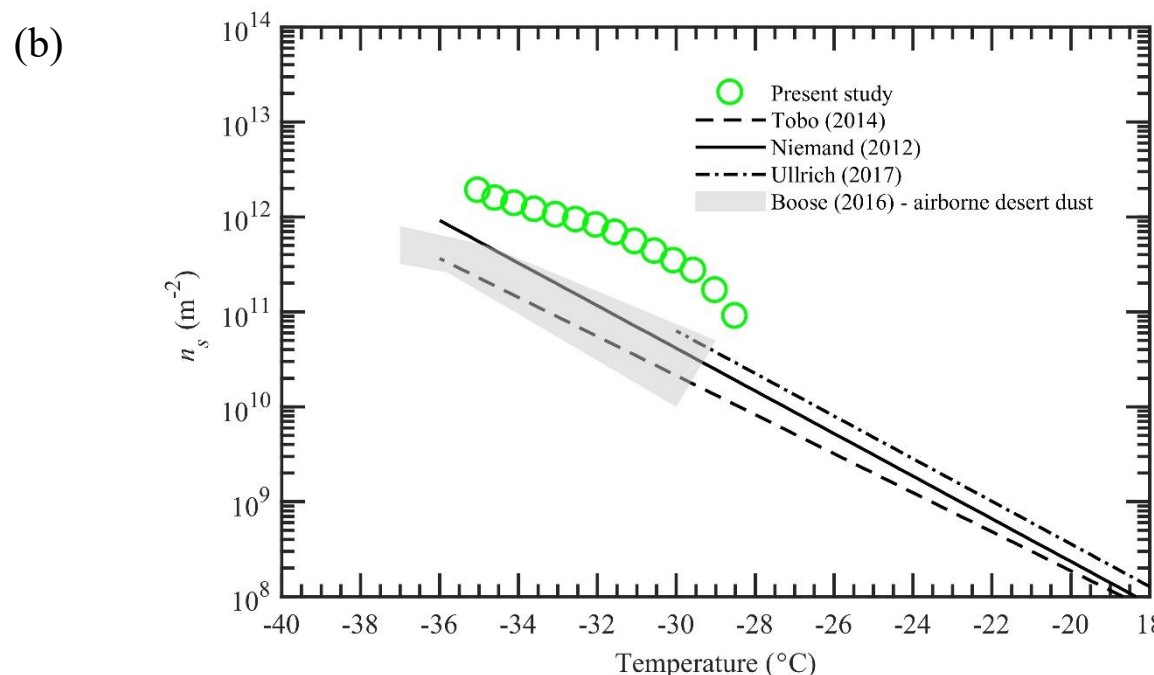




Figure 7: Ice nucleation active site density ($n_s$) as a function of temperature for four INP test species tested in this study. The panels
a) to d) show $n_s$ densities for K-feldspar, airborne arable dust, illite-NX, and Argentinian soil dust, respectively. Solid and dash-dot



lines represent various parameterizations from the literature. See the text for details. Dashed lines in panel a) and d) indicate the extrapolated data calculated outside the temperature limits recommended in these $n_s$ parameterizations. The black color symbols represent $n_s$ values from various other instruments that participated in FIN-02 activity (DeMott et al., 2018). Filled color symbols show the data from the CIC-PNNL chamber but operated at steady-state temperature and $RH_w = 106\%$ conditions at FIN-02. For clarity, confidence intervals are shown only for one data point from each study.