# Peer review of "A new method for operating a continuous flow diffusion chamber to investigate immersion freezing: assessment and performance study"

_Atmospheric Measurement Techniques, 2019_

## Referee Comment (RC1) · Gabor Vali (Referee) · 27 Jan 2020

**Referee comments** on "A new method for operating a continuous flow diffusion chamber to investigate immersion freezing: assessment and performance study" by Gourihar Kulkarni, Naruki Hiranuma, Ottmar Möhler, Kristina Höhler, Swarup China, Daniel 5 J. Cziczo and Paul J. DeMott

**Overview** This paper is a useful addition to the literature on INP measurements in general and to the many reported uses of CFDC instruments in particular. A new mode of operation for a CFDC-type instruments is proposed and evaluated in the paper. In this mode, immersion freezing measurements over a range of temperatures are obtained with steady cooling rather than in the more customary mode of single temperature or step-wise cooling. What is called the evaporation section for many CFDC instruments is changed to nucleation section in this paper .

The proposed method puts emphasis on the temperature dependence of INP activity whereas much of the CFDF literature deals with the dependence of nucleation on humidity, although there is a large range of types and operating modes of CFDC instruments (cf. Hiranuma et al. 2015, with Supplement). The question of the relative importance in these chambers of activation via deposition or freezing is sidestepped in the current paper. It is also set aside in these comments because of the general view that immersion freezing is dominant in most cases.

From an operational point of view, the evaporation of the drops at the low $RH_w$ of the nucleation section avoids the possibility of droplets being counted at the outlet. This avoids one of the common problems with CFDC instruments.

The authors have done a number of tests to support the results presented, and examined some potential error sources. However, probably because the approach is new, additional questions arise and some aspects of the measurement method require further scrutiny.

**Exposure time and temperature** This issue can be addressed principally on the basis of the simulations presented in Section 2.2 of the paper and in the Appendix. According to these calculations droplets rapidly decrease in size at the same time as the temperature adjusts to the temperature of the nucleation chamber, $T_{nc}$. The minimum droplet sizes shown occur when the temperature is within about 1°C of $T_{nc}$. Furthermore, the comparison in Fig. S5 shows that variations in the entry position of the aerosol do not add further errors. From these results it would follow that all droplets reach the set temperature of the nucleation sections within errors comparable to other instrumental uncertainties.

However, the simulations are for ideal laminar flow. To what extent is this actually the case? How much extra spread is caused by deviations from the ideal flow and by polydisperse INP sizes? Larger drops might evaporate later but the temperature they reach would not differ from the set value. But, if there are droplets that evaporate faster than the simulated values, these would have a higher minimum temperature of exposure and that would lead to underestimates of the final results. Since Fig. 7 shows that derived $n_s$ values are higher than those reported in other papers for three out of the four samples tested, it appears that there is no major problem in this reagrd.

More importantly, the short exposure time of INPs to the coldest temperature necessitates consideration of the time dependence of nucleation. Cooling rates of the droplets when entering the nucleation section approach $10°C \ sec^{-1}$ for the lowest $T_{nc}$ value. For such repid cooling, Eq. 5 from Vali and Snider (2015) with $\xi = 0.3$ indicates a 2°C shift toward colder temperatures compared to a 1°C min$^{-1}$ rate of cooling, i.e. the same activity would be observed with 2°C additional cooling[1]. With that correction, the current data in Fig. 7 would have to be represented by points shifted to the right to bring the comparison on the same basis as the other data, although the exact cooling rates associated with each data set from the literature would have to be considered as well.

The tests with constant temperature of the nucleation section (lines 274-281 and solid squares in Fig. 6) do not address the point raised above. This is because the 0.5°C min$^{-1}$ cooling rate is negligible compared to the rapid cooling of the drops on transition from the conditioning to the nucleation section.

While there is no a priori reason for assuming that activity has to rise exponentially, it is also worth considering whether rapid cooling in these experiments may explain why the slopes of the $n_s$ versus $T$ data points in Fig 7 flatten out at colder temperatures. As can be seen from the Figs. S2 to S5, the lower $T_{nc}$ is, the faster the cooling is and thus larger corrections (moving points to higher temperatures) would be necessary to normalize the data to a fixed cooling rate.
* * *
[1]There is no empirical evidence to support the use of the equation for cooling rates 600 times over the reference value, but there is no other basis at this time to make a better estimate.

From the above it follows that the rapid cooling occurring in the transition from the conditioning section to the nucleation section influences both the magnitudes of the derived $n_s$ values and the slopes of the temperature spectra. The authors' view of this would make the paper more complete.

**Sensitivity and error analysis** The paper states that it is possible to execute three test cycles before icing problems. It also states (line 256) that full temperature spectra were acquired in about 30 minutes. However, information about the input aerosol concentrations used in the tests wasn't readily found in the paper. The temperatures of the tests for the airborne dust were restricted to -28°C and colder. It would be useful to know more about sample concentration (in terms of active number at test temperatures), and sampling duration requirements versus statistical counting errors. Perhaps this sort of analysis formed the basis for the accuracy estimates indicated on lines 263-265 of the paper but it is unclear if that is the case.

**Minor points**

line 21 and other places:    Is arable dust a soil science definition?. Perhaps the meaning of the term could be clarified for the context used here. Desert dust? Top soil? Agricultural dust? A detailed description of the sample is given on lines 240 on but the term is used already in the abstract and is frequently used in the paper prior to the definition.

line 62:    "sequence' might be better here than "spectrum".

line 164:    The point about not simulating nucleation is mentioned because of the possible latent heat effect or some other argument?

line 290:    The approximation indicated is valid only for $F_{ice} \ll 1.0$

---

## Referee Comment (RC2) · Anonymous Referee #3 · 8 Mar 2020

**Referee comments on "A new method for operating a continuous flow diffusion chamber to investigate immersion freezing: assessment and performance study" by G. Kulkarni et al. 2020**

In the submitted manuscript Kulkarni et al. describe a new method for operating a Continuous Flow Diffusion Chamber (CFDC) and show both system modeling results and results from testing using various experimental test aerosol and some ambient air sampling. I find the manuscript generally well written and presented. The idea for the new CFDC operation principle is original and enticing. This idea potentially expands the operational range of CFDC instruments and could be a significant contribution to the community. However in its current form the submitted work lacks clarity in some key areas. Some additional work also needs to be done with respect to the figures, where either interpretation is difficult and/or mistakes appear to have been made with labeling in the main text etc.

**Nucleation Temperature and Crystal Growth:** I think the primary question that the authors must clarify is related to quantifying the ice nucleation temperature and ice crystal growth within the evaporation (now nucleation) section of the CFDC. The authors have done a nice job of trying to model the droplet growth in the 'conditioning' section, but have not shown analogous results for modeling the crystal changes in the evaporation section (I recognize they posit that given the saturation condition is $RH_{ice}$ = 100% there are no changes – but consider comment below). My interpretation of Figure 4 and many of the Supplemental figures is that if the system behaves as modeled then liquid droplets quickly evaporate within the nucleation section – on approximately the same time scale as the temperature and RH fields equilibrate. This suggests that the nucleation occurs in this transition region and that the fixed nucleation section temperature in fact controls the gradient between the two sections but does not necessarily represent the actual nucleation condition. What size water droplet must nucleate into ice in order to grow to reach the quoted 3 $\mu$m OPC cutoff for ice? If there is a lower bound on this value then one might interpret before what point along the droplet evaporation curves ice must form. Likewise it would be interesting to understand the range of potential ice crystal sizes depending on at what point entering the chamber a droplet nucleates. Clearly at the warmest temperatures the gradients between the two chambers are weaker and thus the constraints on thermodynamic forcing will be better, but at the colder temperatures I remain to convinced that the nucleation occurs at the equilibrated chamber conditions.

Further evidence already included: on page 6 the authors state that, "ice particle size measured by the OPC can be representative of the size of the droplet while freezing." However, all simulations of droplet growth suggest maximum droplet sizes between 2 and 2.5 $\mu$m. Figure 5 shows peak OPC concentrations from about 3.57 to 5.02 (diameter) which more-or-less corresponds to the peak predicted particle sizes, and those droplet diameters only occur immediately in this transition region and not within the equilibrated portion of the chamber. However, also to consider is that, although the equilibrated chamber represents $RH_{ice}$ = 100%, as long as droplets do exist ice particles can grow due to scavenging...to what extent? Perhaps this is minimal? Will the droplet evaporation go back to the walls?

Below I present an itemized list of additional thoughts and comments as I came to them in the text, which I hope helps to further contextualize my thoughts.

**Itemized Scientific and Editorial Suggestions:**

*Specific Suggestions by Page and Line Number (page, line):*

- (1,26) enough to say sampled from 'am ambient aerosol inlet'. The location etc. is described later.

- (2,42) replace toward with for

- (2,50) percent

- (2,50) CFDCs also

- (2, 61) particles are activated not 'all aerosol'. Remember the strict definition of aerosol is the gas, particle mixture thus activation of all aerosol seems strange.

- (3, 68) the Compact Ice...

• (3, 70) thermally isolated or insulated? How much thermal contact do the 2 sections actually have?

• (3,78) Here begins the use of many symbols $\sim$, $=\sim$, etc. which continues throughout the manuscript in an ill-defined manner. I presume most often these are being used to indicate approximately, for which I suggest $\approx$. Although definitions are a bit muddled the use of similar to $\sim$ to many, including me, denotes an order of magnitude (-ish) approximation. I am sure the authors intention is to convey a more approximate value than that in many of their uses here and throughout.

Here also the RHw is indicated as 106%. Later in the numerical modeling section 2.2 a RHw of 113% is chosen and this value also seems to be chosen in the experimental descriptions that follow. I am left confused, why these differences?

• (3, 80) An OPC

• (3,93) Please also include here the saturation condition that results from the choice of temperatures – it would be nice to also have the value in terms of ice saturation.

• (4, 103) 'The choice of steady-state cooling....' I think the manuscript would benefit from a longer discussion related to the cooling rate. The empirical choice of cooling rate as being satisfactory is supported by the filled symbols on Figure 6, which I understand were measurements made with the chamber at static conditions. However, did the authors try any other cooling rates? Do they have any evidence of what a maximum cooling rate might be? I think any additional information that might have been gathered with regard to the operational limits would add value to what the authors have done.

• (4,110-113) More clarity is needed with respect to the pulse experiments. The pulse duration is quoted as 10.5 s. In Fig. 1 of the supplement the dashed line is used to indicate the limit after which particles are considered to be outside of the lamina. However, if the pulse duration is 10.5 s and the residence time is $\approx 10$s shouldn't pulsed particles continue to arrive until 20.5 s? This would presumably significantly alter the 16% number in the text. How is my understanding deficient?

• (4,113-116) Final sentence of this section seems to be better suited to introduce the following section.

• (5,133-134) such a geometry; I am confused by the end to this sentence. "...it was coupled with energy and viscous heating to enable the species..." I think this needs to be reworded. What was coupled exactly? Is energy conservation meant? Please clarify this sentence.

• (5,141) I found the relevant information is S.1 not S1, but this appears very far into the supplement. It would be useful to order the supplement in an order that corresponds to how it is referenced in the text.

*More notes with regard to S.1:* What is meant with $e_\infty$ and $e_r$? The use of 'environment' is confusing. I think $e_\infty$ represents the far field vapor pressure, while $e_r$ represents the equilibrium vapor pressure at the surface. Similarly the temperature terms should be precisely defined. Furthermore, the $D_v$ term introduces another temperature $T$ and pressure $p$ that seem to have the same definitions as $T_\infty$ and $e_\infty$. Please use uniform notation and be clear.

Finally $r_0$ is the initial radius of the droplet, but by my reading, for the purposes of this manuscript $r_0$ has been set to equal the dry aerosol particle diameter. However, we know that at deliquescence (DRH) any soluble aerosol particle will have a sharp transition terms of growth factor (GF). For example at DRH the GF for NaCl jumps suddenly from 1 to $\approx 1.6$[1]. How is this discontinuity accounted for? Even for mineral surfaces one would expect the $r_0$ to be potentially, importantly different when it is completely coated in bulk water versus when it is dry or just has adsorbed water present.

• (5,143) Figure 2 is referred to but I believe the intent is Figure 4a perhaps?

• (5, 154) Figure S2-5: I found myself spending a lot of time digesting these figures and wonder if the authors should revisit what in fact is best to include in the main text. Perhaps they might hybridize some current figures to add some detail to the main text that only appears now in the supplement. I would also suggest that in Figures S2-S4 the authors choose different color maps for time and RH. With 2 color maps and an offset perhaps panels b and c could potentially be combined. Even if not flipping between figures would be easier if the color maps differed. Figure S5 is missing a legend. Also

in this figure the red droplet radius points seem problematic. Firstly, they seem to show a discontinuity at the chamber transition that none of the other curves indicate. Second, one would intuitively expect their values to perhaps lie between the black and pink, but also the red temperature seems to be lower than the black as it gets close to the transition. Why does the particle further from the cold wall have a colder temperature than that which is closer? I find that a clear explanation of this figure, and especially the reason the red points stand out is lacking.

- (6, 167) Table S1: replace very small with $\leq X$.

- (6, 169) evaporating droplet

- (6, 175) 200 nm? Why not use 300 nm to match the simulations? Perhaps a comment on this choice would be useful.

- (6,177) space betwen RHw and conditions

- (6,180s) See my comment above with regard to the OPC spectra and interpretation based on water droplet size predictions. Also, as a reader it became confusing that the authors switched from discussing droplet radius to droplet diameter when they begin discussing OPC data. I suggest that one dimension is chosen and all discussions and figures converted to this for consistency.

- (6, 183) How are ice particles of 2.0 $\mu$m observed, when previously it was stated a cutoff of 3 $\mu$m is used to select ice? Is this a result of my radius versus diameter confusion?

- (7, 200) "It can be seen that RHw values close to 113% are required before all the AS particles are activated to droplets." I believe the observation is of ice, not of droplet activation. The DRH of ammonium sulfate is $\approx 82\%$ and weakly dependent on temperature. Thus all ammonium sulfate particles should activate at much lower RH. The value of RH here is what is needed for them to grow to a size, subsequently freeze, and remain big enough to be measured as ice.

- (7, 207) Here again another size 400 nm mobility diameter particle is used, perhaps a word as to why this choice was made, relative to the 200nm or 300 nm used in other contexts in the text?

- (7, 211) 7% and 3%

- (7, 222) was once .....is now

- (7, 225) I suggest the authors stick with SI units – mph to m/s.

- (8, 232) particles were also collected.... Were the same particles collected on the SEM films after the CFDC or was this sampling run in parallel?

- (8, 252) froze at the highest

- (9, 263) See previous comment related to temperature ramping.

- (9, 265) allows a comparison with other....

- (9, 271) But citations in order from earliest to latest.

- (9, 278) Here error in $n_s$ is mentioned but does not lead to any uncertainty plotted in Figure 7. In addition to the error bars plotted from other studies it would be nice to have error bars plotted for this study.

- (11, 339) Perhaps the authors could spend some more time attempting to explain why their results seem to be systematically high relative to the other studies (Figure 7). Are there good physical explanations for this?

- (conclusion) From the conclusions I am missing a discussion of whether other existing CFDCs could employ this technique. What for example are the physical constraints in terms of evaporation section length? Given the published geometries of instruments like ZINC[2], SPIN[3] etc. could these instruments hope to run using the operational mode introduced here? Alternatively, if new chambers were being designed what features should be introduced or geometry utilized to enable operation in both traditional and this new mode? Recommendations to the community would strengthen the paper.

- (Figure 1) Can basic chamber dimensions be included, space appears plentiful.

- (Figure 2) Why include temperature from when initial cooling began? Why not just the shaded region, or shaded plus rewarming?

- (Figure 5a) See previous comment with regard to radius versus diameter. Also, why the arbitrary scale? Is this a result of OPC binning? Can scale be changed to be linear? This plot is very hard to interpret in its current form.

- (Figure 6 caption) Suggest a change in text: Other solid square markers represent data collected when the chamber was operated in a steady-state temperature mode (instead of steady cooling).

**Summary:**

I have enjoyed reading the submitted manuscript and find that this is an intriguing new idea. In order to recommend the manuscript for publication I think the authors need to state more convincingly that they constrain the conditions for the observed nucleation. Furthermore, I think the conclusion would be significantly enhanced by describing whether or not other existing CFDC systems could run or test run such a mode of operation. I also encourage the authors to conduct a round of editing to ferret out small mistakes that I found numerous enough that not all could be included here.

—

[1] Castarède, D. and Thomson, E. S. (2018). A thermodynamic description for the hygroscopic growth of atmospheric aerosol particles. *Atmospheric Chemistry and Physics*, 18(20):14939–14948.

[2] Stetzer, O., Baschek, B., Lueoeond, F., and Lohmann, U. (2008). The zurich ice nucleation chamber (zinc) - a new instrument to investigate atmospheric ice formation. *Aerosol Science and Technology*, 42(1):64–74.

[3] Garimella, S., Kristensen, T. B., Ignatius, K., Welti, A., Voigtländer, J., Kulkarni, G. R., Sagan, F., Kok, G. L., Dorsey, J., Nichman, L., Rothenberg, D. A., Rösch, M., Kirchgäßner, A. C. R., Ladkin, R., Wex, H., Wilson, T. W., Ladino, L. A., Abbatt, J. P. D., Stetzer, O., Lohmann, U., Stratmann, F., and Cziczo, D. J. (2016). The spectrometer for ice nuclei (SPIN): an instrument to investigate ice nucleation. *Atmospheric Measurement Techniques*, 9(7):2781–2795.

---

## Referee Comment (RC3) · Anonymous Referee #1 · 16 Mar 2020

The Kulkarni et al, study describes a newly developed operating procedure for investigating the immersion freezing mechanism using continuous flow diffusion chambers. The new method converts the typical nucleation section of such chambers into a "conditioning" section where the aerosol particles are activated into cloud droplets at a fixed temperature where no freezing is expected. Then the particles transition into the newly dubbed "nucleation section" (formerly known as the evaporation section), which is cooled continuously while maintaining ice saturation. The newly developed technique compares well with previously published immersion freezing methods, although it appears to produce higher frozen fractions (within an order of magnitude) than previously observed for several dust species. I find the new method to be well implemented and a

nice addition to the ice nucleation measurement community. I support this manuscript for publication and have the following comments:

General comments:

The residence time of the instrument is described as ∼10 seconds, yet the actual nucleation section is only half of that. This is not that different from traditional CFDCs, however, when the lifetime of the evaporating droplet in the nucleation section is considered, the nucleation time seems closer to ∼2 seconds (according to the numerical simulations). This should be noted in the text.

Furthermore, when considering that the droplets evaporate so quickly, is it possible to retrieve some information about nucleation rates based on the observed ice crystal sizes as a function of temperature, as was alluded to for the homogeneous freezing experiments?

Throughout the text, the new method was described as "the new method". I think it would be nice if the new technique had a name for easier future reference.

I appreciate that the authors did a thorough evaluation of the instrumental design using CFD and pulse experiments. However, I found the description and justification of the settings used missing, see my comment below.

Although the authors go in depth in their comparison with the dusts tested with previous results, I found the justification for the observed differences to be rather vague. This is especially true when comparing with the observations from the FIN workshop where to my understanding, the same aerosols were being tested at the same time. Therefore it would be nice if the authors expanded on some of the reasoning as to why the results in ns can differ by up to an order of magnitude. For example, is it due to not all particles being activated in other techniques due to lamina issues or perhaps it is due to the conditions that the droplets are evaporating at (warm wall temperature or cold wall temperature) etc.?

Technical and minor comments:

Line 38-39: There is mounting evidence that the traditional view of deposition nucleation, may not be occurring. As referenced in the cited Vali et al., (2015) deposition nucleation has also been referred to as immersion freezing in pores or pored condensation and freezing (Marcolli, 2014). Consider adding pore condensation and freezing as a heterogeneous nucleation mechanism.

Line 53-54: Consider adding Garimella et al., (2017) as a reference as well.

Line 57 and 60-61: Did you test to see if all particles did indeed activate as droplets?

Line 77: Are there two sheath flows of 5 lpm of was the total sheath flow 5 lpm? Please clarify.

Line 78: With such a high supersaturation and the required temperature gradient to achieve this supersaturation, how can you ensure that all particles activated as droplets?

Line 91-93: Here the temperature gradient between the walls is mentioned and the achieved temperature of -20 C is described in the following sentence. However, it may be worthwhile to specify the supersaturation of the conditioning section here as well (113 % RHw?).

Line 99-102: This should be reworded, consider something like: "The isothermal conditions of the nucleation section is maintained at ice saturation and cooled at a steady rate (0.5°C min-1 100) by a separate cooling bath in order to determine the immersion freezing efficiency of INPs as a function of supercooled temperature"

Line 102-103: Why does the experiment proceed so far below the homogeneous freezing temperature?

Lines 110-112: Was there any gradient applied to the conditioning experiment during the pulse experiments? I find this unclear in the text. Furthermore, if a temperature

gradient was applied in the conditioning section, are there any effects from the ice coating/ moisture from the walls on the buoyancy profile of the air in the chamber that are missed by doing the test without an ice coating? Also, are there any impacts on the lamina of the chamber when going from the conditioning section to the nucleation section when there is a temperature gradient of 22 C (-20 to -44 C)?

Line 183: remove "either" before "do"

Line 184-186: Please clarify these sentences. Are the smaller droplets at higher temperatures due to the lower nucleation rate and therefore the droplets evaporate more than at colder temperatures where nucleation is faster?

Line 183: Remove "the" between "of" and "supercooled"

Lines 191-195: seem to be contradicting each other, consider rewording.

Line 200-224: Consider breaking this sentence in two for easier readability.

Line 206: Rather than stating "a new mode" perhaps consider stating that it is operated in this specific mode (name the mode).

Line 223-224: Consider rewording.

Line 245-246: Earlier, it is stated that an experiment ends at -44 C yet now the experiment ends at -38, which makes more sense, be sure to be consistent.

References

Garimella, S., Rothenberg, D. A., Wolf, M. J., David, R. O., Kanji, Z. A., Wang, C., Rösch, M. and Cziczo, D. J.: Uncertainty in counting ice nucleating particles with continuous flow diffusion chambers, Atmos Chem Phys, 17(17), 10855–10864, doi:10.5194/acp-17-10855-2017, 2017. Marcolli, C.: Deposition nucleation viewed as homogeneous or immersion freezing in pores and cavities, Atmos Chem Phys, 14(4), 2071–2104, doi:10.5194/acp-14-2071-2014, 2014. Vali, G., DeMott, P. J., Möhler, O. and Whale, T. F.: Technical Note: A proposal for ice nucleation terminology, Atmospheric Chem. Phys., 15(18), 10263–10270, doi:10.5194/acp-15-10263-2015, 2015.

---

## Author Comment (AC1) · 30 Oct 2020

Dear Reviewer,

Thanks for providing these comments to further improve the manuscript. Apologies for the delayed response, the last few months have been challenging during this pandemic. Please find below the reply to your comments. These comments are also used to revise the manuscript.

Thanks,

Gourihar Kulkarni

**Anonymous Referee #RC1**

Referee comments on "A new method for operating a continuous flow diffusion chamber to investigate immersion freezing: assessment and performance study" by Gourihar Kulkarni, Naruki Hiranuma, Ottmar Möhler, Kristina Höhler, Swarup China, Daniel J. Cziczo and Paul J. DeMott

**Overview:**

This paper is a useful addition to the literature on INP measurements in general and to the many reported uses of CFDC instruments in particular. A new mode of operation for a CFDC-type instruments is proposed and evaluated in the paper. In this mode, immersion freezing measurements over a range of temperatures are obtained with steady cooling rather than in the more customary mode of single temperature or step-wise cooling. What is called the evaporation section for many CFDC instruments is changed to nucleation section in this paper.

The proposed method puts emphasis on the temperature dependence of INP activity whereas much of the CFDF literature deals with the dependence of nucleation on humidity, although there is a large range of types and operating modes of CFDC instruments (cf. Hiranuma et al. 2015, with Supplement). The question of the relative importance in these chambers of activation via deposition or freezing is sidestepped in the current paper. It is also set aside in these comments because of the general view that immersion freezing is dominant in most cases.

From an operational point of view, the evaporation of the drops at the low RHw of the nucleation section avoids the possibility of droplets being counted at the outlet. This avoids one of the common problems with CFDC instruments.

The authors have done a number of tests to support the results presented and examined some potential error sources. However, probably because the approach is new, additional questions arise and some aspects of the measurement method require further scrutiny.

Reply: Thanks for the reviews and feedback.

**Exposure time and temperature:**

This issue can be addressed principally on the basis of the simulations presented in Section 2.2 of the paper and in the Appendix. According to these calculations droplets rapidly decrease in size at the same time as the temperature adjusts to the temperature of the nucleation chamber, Tnc. The minimum droplet sizes shown occur when the temperature is within about 1 C of Tnc. Furthermore, the comparison in Fig.

S5 shows that variations in the entry position of the aerosol do not add further errors. From these results it would follow that all droplets reach the set temperature of the nucleation sections within errors comparable to other instrumental uncertainties.

However, the simulations are for ideal laminar flow. **To what extent is this actually the case? How much extra spread is caused by deviations from the ideal flow and by polydisperse INP sizes?** Larger drops might evaporate later but the temperature they reach would not differ from the set value. But, if there are droplets that evaporate faster than the simulated values, these would have a higher minimum temperature of exposure and that would lead to underestimates of the final results. Since Fig. 7 shows that derived ns values are higher than those reported in other papers for three out of the four samples tested, it appears that there is no major problem in this regard.

**Reply**: Agree, these simulations are for ideal laminar flow. To quantify the particles outside of the lamina, pulse test experiments using monodisperse particles are performed. These results are presented in Figure S1 in the original manuscript.

The following text and figure are added to the revised manuscript.

Section 2.1: *Simulations (see below) are performed to investigate the sensitivity of polydisperse particles. The particle residence time of three different monodisperse particles (0.3 µm, 1.0 µm, and 2.0 µm) traversing the chamber is calculated. Results show that the residence time of these particles is similar indicating monodisperse size pulse experiments are also applicable to other size particles.*

[Figure]

*Figure S1: Particle residence time of different size droplets within the chamber.*

More importantly, the short exposure time of INPs to the coldest temperature necessitates consideration of the time dependence of nucleation. Cooling rates of the droplets when entering the nucleation section approach 10°C sec$^{-1}$ for the lowest Tnc value. For such rapid cooling, Eq. 5 from Vali and Snider (2015) with ξ = 0.3 indicates a 2°C shift toward colder temperatures compared to a 1°C min$^{-1}$ rate of cooling, i.e. the same activity would be observed with 2°C additional cooling[1]. With that correction, **the current data in Fig. 7 would have to be represented by points shifted to the right** to bring the comparison on the same basis as the other data, although the exact cooling rates associated with each data set from the literature would have to be considered as well.

The tests with constant temperature of the nucleation section (lines 274-281 and solid squares in Fig. 6) do not address the point raised above. This is because the 0.5° C min[-1] cooling rate is negligible compared to the rapid cooling of the drops on transition from the conditioning to the nucleation section.

While there is no a priori reason for assuming that activity has to rise exponentially, it is also worth considering **whether rapid cooling in these experiments may explain why** the slopes of the ns versus T data points in Fig 7 flatten out at colder temperatures. As can be seen from the Figs. S2 to S5, the lower Tnc is, the faster the cooling is and thus **larger corrections (moving points to higher temperatures) would be necessary to normalize the data** to a fixed cooling rate.

[1]There is no empirical evidence to support the use of the equation for cooling rates 600 times over the reference value, but there is no other basis at this time to make a better estimate.

From the above it follows that the rapid cooling occurring in the transition from the conditioning section to the nucleation section **influences** both the magnitudes of the derived ns values and the slopes of the temperature spectra. **The authors' view of this would make the paper more complete**.

**Reply**: We appreciate these comments. Previously, time-dependent immersion freezing framework (e.g. Vali and Snider (2015), Herbert et al. (2014)) had been formulated to understand time dependent nature of ice nucleation. The framework allows to correct the shift in temperature towards colder temperature for a given change in cooling rate. The cooling rate constant $\xi$ depends on the nature of the INP population, and this constant varies from ~0.15 to 1.6 (Table 2, Herbert et al. (2014)). In this work, we investigated various test species (K-feldspar, airborne soil dusts from the arable region, illite-NX, and Argentinian soil dust), and the $\xi$ values for each of these species for the droplet conditions (size and one INP per droplet) that are used in this work are unknown. Therefore, the application of such an empirical relationship to correct for the shift in temperature because of the droplet cooling rate is not possible currently. These parameters can be obtained by conducting immersion freezing tests using direct processing (e.g. CFDC style instruments) and post-processing (e.g. BINARY style instrument) in parallel.

It should be noted that our experiments shown as solid squares in Fig. 6 do indicate the minimal influence of rapid cooling on ice fraction or $n_s$ values. In this experiment, the nucleation section is held constant at one temperature, and while the droplets are transitioning from the conditioning section to the nucleation section, the droplets undergo rapid cooling. Please see the discussion in section 3.

We added the following paragraph to acknowledge the possibility of influence on reported temperatures because of rapid cooling.

Section 3: *Further, the time-dependent immersion freezing framework (e.g. Vali and Snider (2015), Herbert et al. (2014)) suggests that the rapid cooling of the droplets could shift the cumulative ice fraction towards the colder temperature based on the cooling rate and particular INP material constant. However, these input parameters for the time-dependent model are not available currently to quantify the temperature shift for the present experimental conditions. Future studies that involves collocated direct and post-processing INP instruments would be needed.*

**Sensitivity and error analysis:**

The paper states that it is possible to execute three test cycles before icing problems. It also states (line 256) that full temperature spectra were acquired in about 30 minutes. However, information about the **input aerosol concentrations** used in the tests wasn't readily found in the paper. The temperatures of the

tests for the airborne dust were restricted to -28°C and colder. It would be useful **to know more about sample concentration** (in terms of active number at test temperatures), and **sampling duration requirements versus statistical counting errors**. Perhaps this sort of analysis formed the basis for the accuracy estimates indicated on lines 263-265 of the paper but it is unclear if that is the case.
**Reply**: We added the following sentence.

Section 2.4: *The input aerosol concentration of all four INP species varied from 100 to 800 # per cubic centimeters, and the sampling duration was ~30 minutes.*

The statistical counting error is considered by calculating the standard deviation of the ice fraction measurements. This is discussed in the original manuscript on line 247.

**Minor points:**

line 21 and other places: Is arable dust a soil science definition? Perhaps the meaning of the term could be clarified for the context used here. Desert dust? Top soil? Agricultural dust? A detailed description of the sample is given on lines 240 on but the term is used already in the abstract and is frequently used in the paper prior to the definition.
**Reply**: We modified the definition. The revised definition (in bold) reads as follows.

Abstract: *The performance of the MCIC was evaluated using four INP species: K-feldspar, illite-NX, Argentinian soil dust, and **airborne soil dusts from an arable region** that had shown ice nucleation over a wide span of supercooled temperatures.*

Abstract: *… during the second phase of the Fifth International Ice Nucleation Workshop (FIN-02) campaign, and **airborne arable soil dust** particles were sampled…*

Section 2.4: The arable soil dust is defined as follows.

***Airborne soil dust from the arable region** or shortly airborne arable dust particles* were sampled at the PNNL sampling site during a regional windblown dust event.

line 62: "sequence' might be better here than "spectrum".
**Reply**: Corrected.

line 164: The point about not simulating nucleation is mentioned because of the possible latent heat effect or some other argument?
**Reply**: CFD simulations described in section 2.2 do not involve droplet freezing (or nucleation of ice) simulations. This is not performed because the objective of numerical simulations was to better understand the flow behavior and their impact on droplet dynamics (growth and evaporation). The following sentence added to section 2.2 clarifies the goal of this section.

Section 2.2: *At the entrance of the nucleation section, the temperature and RHw profiles can be unsteady, and to better understand the flow patterns of these profiles within the transitioning zone, and its impact on droplet behavior, numerical simulations using computational fluid dynamics (CFD) are performed.*

line 290: The approximation indicated is valid only for Fice << 1:0
**Reply**: Correct. This assumption of approximation is mentioned in the main text.

Section 3: *The approximation is valid for ice fraction << 1.0.*

---

## Author Comment (AC2) · 30 Oct 2020

**Dear Reviewer,**

**Thanks for providing these comments to further improve the manuscript. Apologies for the delayed response, the last few months have been challenging during this pandemic. Please find below the reply to your comments. These comments are also used to revise the manuscript.**

**Thanks,**

**Gourihar Kulkarni**

**Anonymous Referee #RC2**

Referee comments on "A new method for operating a continuous flow diffusion chamber to investigate immersion freezing: assessment and performance study" by G. Kulkarni et al. 2020

In the submitted manuscript Kulkarni et al. describe a new method for operating a Continuous Flow Diffusion Chamber (CFDC) and show both system modeling results and results from testing using various experimental test aerosol and some ambient air sampling. I find the manuscript generally well written and presented. The idea for the new CFDC operation principle is original and enticing. This idea potentially expands the operational range of CFDC instruments and could be a significant contribution to the community. However in its current form the submitted work lacks clarity in some key areas. Some additional work also needs to be done with respect to the figures, where either interpretation is difficult and/or mistakes appear to have been made with labeling in the main text etc

**Nucleation Temperature and Crystal Growth:** I think the primary question that the authors must clarify is related to quantifying the ice nucleation temperature and ice crystal growth within the evaporation (now nucleation) section of the CFDC. The authors have done a nice job of trying to model the droplet growth in the 'conditioning' section, but have **not shown analogous results for modeling the crystal changes in the evaporation section** (I recognize they posit that given the saturation condition is $RH_{ice}$ = 100% there are no changes – but consider comment below). My interpretation of Figure 4 and many of the Supplemental figures is that if the system behaves as modeled then liquid droplets quickly evaporate within the nucleation section – on approximately the same time scale as the temperature and RH fields equilibrate. This suggests that the nucleation occurs in this transition region and that the fixed nucleation section temperature in fact controls the gradient between the two sections **but does not necessarily represent the actual nucleation condition**. **What size water droplet must nucleate into ice in order to grow to reach the quoted 3 _m OPC cutoff for ice?** If there is a lower bound on this value then one might interpret before **what point along the droplet evaporation curves ice must form**. Likewise it would be interesting to **understand the range of potential ice crystal sizes** depending on at what point entering the chamber a droplet nucleates. Clearly at the warmest temperatures the gradients between the two chambers are weaker and thus the constraints on thermodynamic forcing will be better, but at the colder **temperatures I remain to convinced that the nucleation occurs at the equilibrated chamber conditions.**

Reply: The evaporation section conditions are constant. This section is maintained at constant temperature and $RH_{ice}$ (=100%) conditions (Fig. 4a). We expect no change in the ice crystal size.

Correct, the temperature within the transition section (varies from conditioning section to nucleation or evaporation section) does not correspond to the equilibrium nucleation section temperature (e.g. Fig. S5

b). Freezing occurs at various temperatures that range from the conditioning section temperature (~ -20 °C) to the nucleation section temperature (e.g. -30°C) conditions. The temperature uncertainty across the aerosol lamina and nucleation section are $\pm 0.9$ and $\pm 0.4$°C, respectively. Here, we have used temperature uncertainty across the nucleation section as the temperature uncertainty within the ice fraction. The ice fraction is defined as the cumulative fraction of the droplet frozen, and it is reported at the coldest section of the chamber (i.e. steady state nucleation section temperature). See supplementary section Text S1. Following sentence is added.

Text S1: *The freezing temperature (T) is defined at the steady state temperature of the nucleation section, and the freezing temperature uncertainty is assumed to be similar to the uncertainty across the nucleation section ($\pm 0.4$°C).*

CFD simulations (e.g. Fig S3 c) show that water droplets of size greater than 2 μm in radius will mostly contribute towards nucleation of ice. Droplets smaller than this size are exposed to subsaturation conditions, and they evaporate quickly (< 1 sec; see Fig S3 b). It should be noted that as nucleation occurs in the order of a few ms (Holden et al. 2019), the droplets smaller than 2 μm might also contribute towards nucleation of ice. However, the contribution of these smaller droplets of less than 2 μm is very small (see Fig. 5a).

*Holden, M. A., Whale, T. F., Tarn, M. D., O'Sullivan, D., Walshaw, R. D., Murray, B. J., Meldrum, F. C., and Christenson, H. K.: High-speed imaging of ice nucleation in water proves the existence of active sites, Sci. Adv., 5, eaav4316, https://doi.org/10.1126/sciadv.aav4316, 2019.*

Further evidence already included: on page 6 the authors state that, "ice particle size measured by the OPC can be representative of the size of the droplet while freezing." However, all simulations of droplet growth suggest maximum droplet sizes between 2 and 2.5 _m. Figure 5 shows peak OPC concentrations from about 3.57 to 5.02 (diameter) which more-or-less corresponds to the peak predicted particle sizes, and those droplet diameters only occur immediately in this transition region and **not within the equilibrated portion of the chamber**. However, also to consider is that, although the equilibrated chamber represents RH$_{ice}$ = 100%, as long as **droplets do exist ice particles can grow due to scavenging...to what extent? Perhaps this is minimal? Will the droplet evaporation go back to the walls?**

Reply: Figure 5a shows ice crystal sizes and their respective concentrations at different temperatures. As mentioned above, droplets of size less than 2 μm in radius may contribute towards the total ice crystal concentration, but their fraction compared to the total concentration is very small.

Flow conditions across the chamber are laminar (Fig. 4a). The droplets and ice crystals follow particle trajectories determined by the various forces (flow conditions and gravity) acting on the particle. It appears that these particles have insufficient inertia to cross the gas streamlines (Fig. S5; see five INP trajectories), such that scavenging of droplets by ice crystals can be ignored. Correct, the water vapor from the droplet (during evaporation) might go towards the wall. Also, some of the vapor might exit the chamber.

Below I present an itemized list of additional thoughts and comments as I came to them in the text, which I hope helps to further contextualize my thoughts.

**Itemized Scientific and Editorial Suggestions:**
Specific Suggestions by Page and Line Number (page, line):

_ (1,26) enough to say sampled from 'am ambient aerosol inlet'. The location etc. is described later.
_ (2,42) replace toward with for
_ (2,50) percent
_ (2,50) CFDCs also
_ (2, 61) particles are activated not 'all aerosol'. Remember the strict definition of aerosol is the gas, particle mixture thus activation of all aerosol seems strange.
_ (3, 68) the Compact Ice...
**Reply**: All the above comments are addressed.

_ (3, 70) thermally isolated or insulated? How much thermal contact do the 2 sections actually have?
**Reply**: Corrected, they are thermally isolated. The two walls are not in contact with each other, but they are separated by double-layered insulated gasket.

_ (3,78) Here begins the use of many symbols _, =_, etc. which continues throughout the manuscript in an ill-defined manner. I presume most often these are being used to indicate approximately, for which I suggest _. Although definitions are a bit muddled the use of similar to _ to many, including me, denotes an order of magnitude (-ish) approximation. I am sure the authors intention is to convey a more approximate value than that in many of their uses here and throughout.
**Reply**: Corrected. The ~ symbol is replaced with ≈ symbol.

Here also the RHw is indicated as 106%. Later in the numerical modeling section 2.2 a RHw of 113% is chosen and this value also seems to be chosen in the experimental descriptions that follow. I am left confused, why these differences?
**Reply**: The RHw = 106% corresponds to the CIC chamber (the original chamber, but not the modified chamber or MCIC). The RHw = 113% corresponds to the modified CIC chamber (MCIC).

_ (3, 80) An OPC
**Reply**: Corrected.

_ (3,93) Please also include here the saturation condition that results from the choice of temperatures – it would be nice to also have the value in terms of ice saturation.
**Reply**: The saturation (water and ice) conditions for these conditions are shown in Figure 3. We added the following sentence to address this comment.

Section 2.1: *The resulting water and ice saturation conditions are shown in Figure 3.*

_ (4, 103) 'The choice of steady-state cooling....' I think the manuscript would benefit from a longer discussion related to the cooling rate. The empirical choice of cooling rate as being satisfactory is supported by the filled symbols on Figure 6, which I understand were measurements made with the chamber at static conditions. However, did the authors try any other cooling rates? Do they have any evidence of what a maximum cooling rate might be? I think any additional information that might have been gathered with regard to the operational limits would add value to what the authors have done.
**Reply**: Some exploratory work with different cooling rates is explored. The ice fraction results of ambient sampling showed a negligible difference between the cooling rates.

[Figure]

Figure S8: The $F_{ice}$ of airborne arable dust species as a function of temperature and nucleation section cooling rates. The cooling rate of 0.5 Kmin$^{-1}$ was used in this study.

The figure is added to the supplementary section. The following text is added to the manuscript.

Section 2.1: *The implications of higher cooling rates towards INP measurements were also explored.*
Section 3.0: *The experiments with higher cooling rates (2.5 and 7.0 °C min$^{-1}$) had also a negligible effect on $F_{ice}$ of airborne arable dust species (Figure S8).*

_ (4,110-113) More clarity is needed with respect to the pulse experiments. The pulse duration is quoted as 10.5 s. In Fig. 1 of the supplement the dashed line is used to indicate the limit after which particles are considered to be outside of the lamina. However, if the pulse duration is 10.5 s and the residence time is _ 10s shouldn't pulsed particles continue to arrive until 20.5 s? This would presumably significantly alter the 16% number in the text. How is my understanding deficient?
**Reply**: The data is shown when CPC starts recording the particle counts. The below sentence is added to the figure caption of figure S1.

Figure S1*: The data is shown only when CPC started recording the particle counts.*

_ (4,113-116) Final sentence of this section seems to be better suited to introduce the following section.
**Reply**: Corrected.

_ (5,133-134) such a geometry; I am confused by the end to this sentence. "…it was coupled with energy and viscous heating to enable the species…" I think this needs to be reworded. What was coupled exactly? Is energy conservation meant? Please clarify this sentence.
**Reply**: Corrected. These sentences describe the viscous model used to model the flow and droplet trajectories. The sentence is revised as follows.

Section 2.2: *The viscous model – the standard RNG $\kappa - \varepsilon$ turbulence model was used. This model treats velocity fluctuations better than other turbulence models for such a geometry. This turbulence model was used in conjunction with species transport modeling capability such that the effects of smaller eddies of fluid motion are better captured.*

_ (5,141) I found the relevant information is S.1 not S1, but this appears very far into the supplement. It would be useful to order the supplement in an order that corresponds to how it is referenced in the text.
**Reply**: Sorry for the inconvenience. It should read S.1. To avoid the confusion, we rename it as Text S1.

The order of Text S1 and S2 is rearranged.

More notes with regard to S.1: What is meant with $e_1$ and $e_r$? The use of 'environment' is confusing. I think $e_1$ represents the far field vapor pressure, while $e_r$ represents the equilibrium vapor pressure at the surface. Similarly the temperature terms should be precisely defined. Furthermore, the $D_v$ term introduces another temperature T and pressure p that seem to have the same definitions as $T_1$ and $e_1$. Please use uniform notation and be clear.
**Reply**: Corrected.

Finally $r_0$ is the initial radius of the droplet, but by my reading, for the purposes of this manuscript $r_0$ has been set to equal the dry aerosol particle diameter. However, we know that at deliquescence (DRH) any soluble aerosol particle will have a sharp transition terms of growth factor (GF). For example at DRH the GF for NaCl jumps suddenly from 1 to _ $1:6_1$. How is this discontinuity accounted for? Even for mineral surfaces one would expect the $r_0$ to be potentially, importantly different when it is completely coated in bulk water versus when it is dry or just has adsorbed water present.
**Reply**: The $r_0$ sizes are already CCN sizes. We repeat the sentence already described in section 2.2.

The potential INPs are assumed (i.e., sub-saturated particle growth is ignored) to activate to droplets because they are greater than cloud condensation nuclei sizes (Seinfeld and Pandis, 2016) and grow as long as RHw is increasing or remains constant.

_ (5,143) Figure 2 is referred to but I believe the intent is Figure 4a perhaps?
**Reply**: Thanks. Yes, it is Figure 4a. The typo is corrected.

_ (5, 154) Figure S2-5: I found myself spending a lot of time digesting these figures and wonder if the authors should revisit what in fact **is best to include in the main text**. Perhaps they **might hybridize some current figures to add some detail to the main text** that only appears now in the supplement.
**Reply**: An example is already included in the main text. See Figure 4b. A reference to other supplementary figures is included in the figure caption

I would also suggest that in Figures S2-S4 the authors choose different color maps for time and RH.
**Reply**: Thanks for the suggestion. This is tried but gets overly complicated to interpret the results. The choice of similar colormap is justified because then it is easy to compare the low and high values using consistent colors.

With 2 color maps and an offset perhaps panels b and c could potentially be combined. Even if not flipping between figures would be easier if the color maps differed.

**Reply**: Addressed above.

Figure S5 is missing a legend. Also in this figure the red droplet radius points seem problematic. Firstly, they seem to show a discontinuity at the chamber transition that none of the other curves indicate.

Second, one would intuitively expect their values to perhaps lie between the black and pink, but also the red temperature seems to be lower than the black as it gets close to the transition. Why does the particle further from the cold wall have a colder temperature than that which is closer? I find that a clear explanation of this figure, and especially the reason the red points stand out is lacking.
**Reply**: The legend is like in Figure S4. The following sentence is added to the caption.

Figure S5: *The plotted data line style and marker symbol are similar to the legend described in Figure S4.*

_ (6, 167) Table S1: replace very small with _ X.
**Reply**: Corrected.

_ (6, 169) evaporating droplet
**Reply**: Corrected.

_ (6, 175) 200 nm? Why not use 300 nm to match the simulations? Perhaps a comment on this choice would be useful.
**Reply**: The choice was based on the optimization of two factors: number concentration and monodisperse size. This size allowed us to generate the maximum number of monodisperse particles. Generating smaller sizes produces multiple charge particles, whereas generating larger size particles produces fewer particles. The following sentence is added.

Section 2.3: *The choice of this size allowed us to generate the maximum number concentration of monodisperse particles.*

_ (6,177) space betwen RHw and conditions
**Reply**: Corrected.

_ (6,180s) **See my comment** above with regard to the **OPC spectra and interpretation** based on water droplet size predictions. Also, as a reader it became confusing that the authors switched from discussing droplet radius to droplet diameter when they begin discussing OPC data. **I suggest** that one dimension is chosen and all discussions and figures converted to this for consistency.
**Reply**: The comments regarding ice crystal size and the relationship between droplet and ice crystal size related to the nucleation section temperature are addressed above.
We revised Figure 5a such that Yaxis shows the particle units in radius, and it is now consistent with the other figures.

*The following figure is added.*

[Figure]

**Figure 5: Homogenous freezing of water droplets containing one wt. % ammonium sulfate solution. (a) OPC classified ice particle concentrations as a function of ice crystal diameter at different temperatures. Warm and cold walls of the conditioning section are maintained at -9 and -27°C, respectively.**

_ (6, 183) How are ice particles of 2.0 _m observed, when previously it was stated a cutoff of 3 _m is used to select ice? Is this a result of my radius versus diameter confusion?
**Reply**: The cutoff of 3 µm is defined in diameter. The new sentence in section 2.1 reads as

Section 2.1: *… certain size-threshold (≈3 µm **in diameter**).*

We also revised the cut size definition in section 2.3. The new sentence in section 2.3 reads as

Section 2.3: *… we observe ice particles of size ≈ 2.0 µm **in diameter**.*

_ (7, 200) "It can be seen that RHw values close to 113% are required before all the AS particles are activated to droplets." I believe the observation is of ice, not of droplet activation. The DRH of ammonium sulfate is _ 82% and weakly dependent on temperature. Thus all ammonium sulfate particles should activate at much lower RH. The value of RH here is what is needed for them to grow to a size, subsequently freeze, and remain big enough to be measured as ice.
**Reply**: Agree. The sentence is revised as follows. The original sentences that follow this sentence discuss the importance of high RHw conditions.

Section 2.3: *It can be seen that RHw values close to 113% are required before all the AS particles are activated to droplets **and measured as ice crystals** (Figure 5b).*

_ (7, 207) Here again another size 400 nm mobility diameter particle is used, perhaps a word as to why this choice was made, relative to the 200nm or 300 nm used in other contexts in the text?
**Reply**: Following words to the existing sentence are added.

Section 2.4: Laboratory measurements showed that the contribution of double and triple charged particles was less than 7 and 3%, respectively, **which also justified the choice of 400 nm size particles.**

_ (7, 211) 7% and 3%

**Reply**: Corrected.

_ (7, 222) was once .....is now
**Reply**: Corrected.

_ (7, 225) I suggest the authors stick with SI units – mph to m/s.
**Reply**: Corrected.

_ (8, 232) particles were also collected.... Were the same particles collected on the SEM films after the CFDC or was this sampling run in parallel?
**Reply**: It was run in parallel. Highlighted words are added, and the existing sentence is revised as follows.

Section 2.4: **In parallel to INP measurements,** the particles were collected on a carbon type-B film (Ted Pella Inc.; 01814-F) for scanning electron microscopy-energy dispersive x-ray spectroscopy (SEM-EDS) analysis to better understand the size distribution and composition of these airborne dust particles.

_ (8, 252) froze at the highest
**Reply**: Corrected.

_ (9, 263) See previous comment related to temperature ramping.
**Reply**: This comment is addressed above. See new figure Figures S8 caption.

Figure S8: *The $F_{ice}$ of airborne arable dust species as a function of temperature and nucleation section cooling rates. The cooling rate of 0.5 Kmin$^{-1}$ was used in this study.*

_ (9, 265) allows a comparison with other....
**Reply**: Corrected.

_ (9, 271) But citations in order from earliest to latest.
**Reply**: Corrected.

_ (9, 278) Here error in $n_s$ is mentioned but does not lead to any uncertainty plotted in Figure 7. In addition to the error bars plotted from other studies it would be nice to have error bars plotted for this study.
**Reply**: The errors are plotted but they are invisible in the figure. For example, for K-Feldspar species, the $n_s$ value at -22°C is 0.1083x10$^{12}$ (m$^{-2}$) and the error is 1.613x10$^7$ (m$^{-2}$).

_ (11, 339) Perhaps the authors could spend some more time attempting to explain why their results seem to be systematically high relative to the other studies (Figure 7). Are there good physical explanations for this?
**Reply**: In addition to the different measurement methods that might have led to this discrepancy (already discussed in the main paper); it is also possible the experimental uncertainties from different ns parameters (e.g. ice crystal detection limit, RH, and temperature error limits) could also influence the ns calculations. The following sentence is added.

Section 3: *The experimental uncertainties (e.g. ice crystal detection limit, RH, and temperature error limits) from these methods could also influence the $n_s$ results.*

_ (conclusion) From the conclusions I am missing a discussion of whether other existing CFDCs could employ this technique. What for example are the physical constraints in terms of evaporation section length? Given the published geometries of instruments like ZINC2, SPIN3 etc. could these instruments hope to run using the operational mode introduced here? Alternatively, if new chambers were being designed what features should be introduced or geometry utilized to enable operation in both traditional and this new mode? Recommendations to the community would strengthen the paper.

**Reply**: Yes, other CFDC's could employ this new method. Based on CFD results (Fig.S3-5), the minimum evaporation/nucleation length required is 0.2 m. Implementing a separate refrigeration system to independently cool the nucleation section, the new operation mode can be adapted. For a new chamber geometry, the length of the conditioning section can be increased such that droplet size can be increased. This feature is useful such that the lifetime of the ice layer can be increased because higher RHw = 113% is not needed.

The following sentences are added to section 3.

Section 3: *Our results can guide design considerations for future CFDC-style ice chambers. The length of the conditioning section can be increased so that higher RHw would not be necessary to activate all the particles to sufficiently large droplet sizes (≈ 2 μm in diameter). This design feature could help to increase the lifetime of the ice layer. Based on CFD results (Fig.S3-5), the minimum evaporation/nucleation length required is 0.2 m. Also, implementing a separate refrigeration system to independently cool the nucleation section, the presented new operation mode can be adapted.*

_ (Figure 1) Can basic chamber dimensions be included, space appears plentiful.
**Reply**: We added the following sentence to the figure caption of Figure 1.

Figure 1: *The length of both the conditioning and nucleation section is 0.45 m. The width of the chamber is 0.15 m. The gap between warm and cold walls is 0.01 m.*

_ (Figure 2) Why include temperature from when initial cooling began? Why not just the shaded region, or shaded plus rewarming?
**Reply**: This is shown to give an idea of temperature time-series from the beginning of the experiment.

_ (Figure 5a) See previous comment with regard to radius versus diameter. Also, why the arbitrary scale? Is this a result of OPC binning? Can scale be changed to be linear? This plot is very hard to interpret in its current form.
**Reply**: Figure 5a is revised, please see above. A new figure is added that shows the Y-axis in particle size in radius units.
The scale is fixed. It was plotting typo. Adopting a linear scale makes the figure difficult to analyze. The new figure is clearer.

_ (Figure 6 caption) Suggest a change in text: Other solid square markers represent data collected when the chamber was operated in a steady-state temperature mode (instead of steady cooling).
**Reply**: Thanks for the suggestion. The sentence is revised.

**Summary:**
I have enjoyed reading the submitted manuscript and find that this is an intriguing new idea. In order to recommend the manuscript for publication I think the authors **need to state more convincingly** that

they constrain the conditions for the observed nucleation. Furthermore, I think the conclusion would be significantly enhanced by describing **whether or not other existing CFDC systems** could run or test run such a mode of operation. I also encourage the authors **to conduct a round of editing** to ferret out small mistakes that I found numerous enough that not all could be included here.

**Reply**: Thanks for these comments. The ice fraction is defined, see Text S1. The design recommendations for future CFDC chamber development are described in section 3. English editing was performed.

—

[1] Castar`ede, D. and Thomson, E. S. (2018). A thermodynamic description for the hygroscopic growth of atmospheric aerosol particles. Atmospheric Chemistry and Physics, 18(20):14939–14948.

[2] Stetzer, O., Baschek, B., Lueoeond, F., and Lohmann, U. (2008). The zurich ice nucleation chamber (zinc) - a new instrument to investigate atmospheric ice formation. Aerosol Science and Technology, 42(1):64–74.

[3] Garimella, S., Kristensen, T. B., Ignatius, K., Welti, A., Voigtl ¨ander, J., Kulkarni, G. R., Sagan, F., Kok, G. L., Dorsey, J., Nichman, L., Rothenberg, D. A., R¨osch, M., Kirchg¨aßner, A. C. R., Ladkin, R., Wex, H., Wilson, T. W., Ladino, L. A., Abbatt, J. P. D., Stetzer, O., Lohmann, U., Stratmann, F., and Cziczo, D. J. (2016). The spectrometer for ice nuclei (SPIN): an instrument to investigate ice nucleation. Atmospheric Measurement Techniques, 9(7):2781–2795.

---

## Author Comment (AC3)

**Dear Reviewer,**

**Thanks for providing these comments to further improve the manuscript. Apologies for the delayed response, the last few months have been challenging during this pandemic. Please find below the reply to your comments. These comments are also used to revise the manuscript.**

**Thanks,**

**Gourihar Kulkarni**

**Anonymous Referee #RC3**

The Kulkarni et al, study describes a newly developed operating procedure for investigating the immersion freezing mechanism using continuous flow diffusion chambers. The new method converts the typical nucleation section of such chambers into a "conditioning" section where the aerosol particles are activated into cloud droplets at a fixed temperature where no freezing is expected. Then the particles transition into the newly dubbed "nucleation section" (formerly known as the evaporation section), which is cooled continuously while maintaining ice saturation. The newly developed technique compares well with previously published immersion freezing methods, although it appears to produce higher frozen fractions (within an order of magnitude) than previously observed for several dust species. I find the new method to be well implemented and a nice addition to the ice nucleation measurement community. I support this manuscript for publication and have the following comments:

General comments:

The residence time of the instrument is described as _10 seconds, yet the actual nucleation section is only half of that. This is not that different from traditional CFDCs, however, when the lifetime of the evaporating droplet in the nucleation section is considered, the nucleation time seems closer to _2 seconds (according to the numerical simulations). This should be noted in the text.

**Reply**: Following sentence is added. The word 'particle' is added to say that total particle residence within the chamber is ~ 10 s.

Section 2.1: …which limits the total **particle** residence time to ≈10 s. The droplet residence and nucleation time within the chamber are a maximum of 6.5 s and 2 s, respectively.

Furthermore, when considering that the droplets evaporate so quickly, is it possible to retrieve some information about nucleation rates based on the observed ice crystal sizes as a function of temperature, as was alluded to for the homogeneous freezing experiments?

**Reply**: This is another way of expressing INP measurements (Herbert et al. 2014). We know the ice fraction and particle surface area; however, nucleation time is uncertain. These inputs can be used to calculate the nucleation rate ($J_{het}$). Alternatively, a normalized freezing rate (R/A) can be calculated. We hope to provide the raw data upon request, and this data information would allow readers to calculate these rates.

Herbert, R. J., Murray, B. J., Whale, T. F., Dobbie, S. J., and Atkinson, J. D.: Representing time-dependent freezing behaviour in immersion mode ice nucleation, Atmos Chem Phys, 14, 8501-8520, 10.5194/acp-14-8501-2014, 2014.

Throughout the text, the new method was described as "the new method". I think it would be nice if the new technique had a name for easier future reference.
**Reply**: We call this new technique as 'Modified Compact Ice Chamber' or 'MCIC.' The manuscript is revised, and sentences are revised to incorporate MCIC.

Section 2.1: *Figure 1 shows a vertical cross-sectional geometry of the modified mode PNNL ice chamber,* *which is now referred to as a Modified Compact Ice Chamber (MCIC).*

Section 2.4: *The immersion freezing efficiency of K-feldspar, illite-NX, Argentinian soil dust, and airborne arable dust particles was measured to test the performance of the **MCIC**.*
Section 3: *A good agreement with the results obtained from **MCIC** was observed, …*

Section 3: *….4 up to 5 is needed to apply to the CIC-PNNL data to match with the data from the **MCIC**.*

Section 4: An alternative method of operating a CFDC-style ice chamber **referred as MCIC** was explored to …

I appreciate that the authors did a thorough evaluation of the instrumental design using CFD and pulse experiments. However, I found the description and justification of the settings used missing, see my comment below.

Although the authors go in depth in their comparison with the dusts tested with previous results, I found the justification for the observed differences to be rather vague. This is especially true when comparing with the observations from the FIN workshop where to my understanding, the same aerosols were being tested at the same time. Therefore it would be nice if the authors expanded on some of the reasoning as to why the results in ns can differ by up to an order of magnitude. For example, is it due to not all particles being activated in other techniques due to lamina issues or perhaps it is due to the
conditions that the droplets are evaporating at (warm wall temperature or cold wall temperature) etc.?
**Reply**: In addition to the different measurement methods that might have led to this discrepancy (already discussed in the main paper); it is also possible the experimental uncertainties from different $n_s$ parameters (e.g. ice crystal detection limit, RH and temperature error limits) could also influence the $n_s$ calculations. Following sentence is added.

Section 3: *The experimental uncertainties (e.g. ice crystal detection limit, RH, and temperature error limits) from these methods could also influence the ns results.*

Technical and minor comments:

Line 38-39: There is mounting evidence that the traditional view of deposition nucleation, may not be occurring. As referenced in the cited Vali et al., (2015) deposition nucleation has also been referred to as immersion freezing in pores or pored condensation and freezing (Marcolli, 2014). Consider adding pore condensation and freezing as a heterogeneous nucleation mechanism.
**Reply**: Following sentence is added.

Section 1: *Deposition nucleation has also been referred to as pore condensation and freezing mechanism because it is similar to as immersion freezing but in pores (Marcolli 2014).*

Line 53-54: Consider adding Garimella et al., (2017) as a reference as well.
**Reply**: Added.

Line 57 and 60-61: Did you test to see if all particles did indeed activate as droplets?
**Reply**: This was tested by freezing the droplets at and below homogeneous freezing temperatures. See Figure 5b.

Line 77: Are there two sheath flows of 5 lpm of was the total sheath flow 5 lpm? Please clarify.
**Reply**: There is one sheath flow. The existing sentence is revised.

Section 2.1: The **single** sheath and sample flow rates were 5 and 1 liters per minute (LPM), respectively, …

Line 78: With such a high supersaturation and the required temperature gradient to achieve this supersaturation, how can you ensure that all particles activated as droplets?
**Reply**: This was tested by freezing the droplets at and below homogeneous freezing temperatures. See Figure 5b.

Line 91-93: Here the temperature gradient between the walls is mentioned and the achieved temperature of -20 C is described in the following sentence. However, it may be worthwhile to specify the supersaturation of the conditioning section here as well (113 % RHw?).
**Reply**: Following sentence is added.

Section 2.1: *The resulting water and ice saturation conditions are shown in Figure 3.*

Line 99-102: This should be reworded, consider something like: "The isothermal conditions of the nucleation section is maintained at ice saturation and cooled at a steady rate (0.5_C min-1 100) by a separate cooling bath in order to determine the immersion freezing efficiency of INPs as a function of supercooled temperature"
**Reply**: Thanks for the suggestion. The sentence is revised as follows.

Section 2.1: *The isothermal conditions of the nucleation section is maintained at ice saturation and cooled at a steady rate (0.5°C min$^{-1}$) by a separate cooling bath to determine the immersion freezing efficiency of INPs as a function of supercooled temperature.*

Line 102-103: Why does the experiment proceed so far below the homogeneous freezing temperature?
**Reply**: The experiment could have terminated at the onset of homogeneous freezing temperature (-38 to -39 °C). Cooling below this temperature allowed us to obtain measurements at homogeneous freezing temperature for ~10 minutes. This additional data helped towards quality control and to account for the uncertainty within the temperature. The following sentence is added.

Section 2.1: *This additional supercooling below the onset of homogeneous freezing temperature allowed to obtain freezing data that was used towards data quality control and to account for the uncertainty within the temperature.*

Lines 110-112: Was there any gradient applied to the conditioning experiment during the pulse experiments? I find this unclear in the text. Furthermore, if a temperature gradient was applied in the conditioning section, are there any effects from the ice coating/ moisture from the walls on the buoyancy

profile of the air in the chamber that are missed by doing the test without an ice coating? Also, are there any impacts on the lamina of the chamber when going from the conditioning section to the nucleation section when there is a temperature gradient of 22 C (-20 to -44 C)?

**Reply:** There was no gradient applied to the conditioning section of the chamber.

Flow conditions across the chamber are laminar (see Fig. 4a). The INP trajectory determined by the various forces (flow conditions and gravity) acting on the particle follows the fluid flow streamlines. Figure 3 shows the steady-state airflow velocity within the conditioning section of the chamber. These results indicate that the chamber conditions do not affect the buoyancy profile of the air. Therefore, particle pulse experiments are also valid after ice coating.

Figures S2-5 show no effect of the temperature gradient between the conditioning and nucleation section temperature on the aerosol lamina within the conditioning section and transitioning zone.

Line 183: remove "either" before "do"
**Reply**: Corrected.

Line 184-186: Please clarify these sentences. Are the smaller droplets at higher temperatures due to the lower nucleation rate and therefore the droplets evaporate more than at colder temperatures where nucleation is faster?

**Reply**: The droplet evaporation is observed from -20 till -37.5°C, see Figures S2 – 4. These figures show that droplets evaporate at the entrance of the conditioning section. E.g. Fig S3 c show that water droplet of size greater than 2 µm in radius will mostly contribute towards nucleation of ice. Droplets smaller than this size are exposed to subsaturation conditions, and they evaporate quickly (< 1 sec; see Fig S3 b). It should be noted that as nucleation occurs in the order of a few ms (Holden et al. 2019), the droplets smaller than 2 µm might also contribute towards nucleation of ice. However, the contribution of these smaller droplets of less than 2 µm is very small (see Fig. 5a).

Line 183: Remove "the" between "of" and "supercooled"
**Reply**: Corrected.

Lines 191-195: seem to be contradicting each other, consider rewording.
**Reply**: The sentences are revised as follows.

Section 2.3: *We find good agreement between the experimental and predicted freezing temperatures. These results also show the complete evaporation of supercooled droplets within the nucleation section, because no ice particles are observed above ~≈ 37.5°C, and therefore the freezing results (see section 3) at warmer temperatures (> -37°C) can be ascribed as the heterogeneous freezing of the droplets or immersion freezing.*

Line 200-224: Consider breaking this sentence in two for easier readability.
**Reply**: The sentence is divided into two sentences for clarity.

Section 2.3: *Higher RHw values enable the encapsulation of all particles that are within and may spread outside (Garimella et al. 2017) the width of aerosol lamina into droplets. **In addition,** high saturation conditions also help to grow the droplets to the larger size; so, they survive long enough to induce the freezing of droplets within the nucleation section.*

Line 206: Rather than stating "a new mode" perhaps consider stating that it is operated in this specific mode (name the mode).

**Reply**: We call this new technique as 'Modified Compact Ice Chamber' or 'MCIC.' The manuscript is revised, and sentences are revised to incorporate MCIC.

Line 223-224: Consider rewording.

**Reply**: The sentence is revised as follows.

Section 2.4: The region was once covered with basalt lava, but is now built up with loose topsoil – loess.

Line 245-246: Earlier, it is stated that an experiment ends at -44 C yet now the experiment ends at -38, which makes more sense, be sure to be consistent.

**Reply**: Sorry for the confusion. Although the experiment ends at -44°C, the INP data from -20 to -38°C is only investigated and presented in this study.

**References**

Garimella, S., Rothenberg, D. A., Wolf, M. J., David, R. O., Kanji, Z. A., Wang, C., Rösch, M. and Cziczo, D. J.: Uncertainty in counting ice nucleating particles with continuous flow diffusion chambers, Atmos Chem Phys, 17(17), 10855–10864, doi:10.5194/acp-17-10855-2017, 2017.

Marcolli, C.: Deposition nucleation viewed as homogeneous or immersion freezing in pores and cavities, Atmos Chem Phys, 14(4),2071–2104, doi:10.5194/acp-14-2071-2014, 2014.

Vali, G., DeMott, P. J., Möhler, O. and Whale, T. F.: Technical Note: A proposal for ice nucleation terminology, Atmospheric Chem. Phys., 15(18), 10263–10270, doi:10.5194/acp-15-10263-2015, 2015.